# STABILIZING BACKPROPAGATION THROUGH TIME TO LEARN COMPLEX PHYSICS

**Patrick Schnell & Nils Thuerey**
School of Computation, Information and Technology
Technical University of Munich
Boltzmannstr. 3, 85748 Garching, Germany
{patrick.schnell,nils.thuerey}@tum.de

## ABSTRACT

Of all the vector fields surrounding the minima of recurrent learning setups, the gradient field with its exploding and vanishing updates appears a poor choice for optimization, offering little beyond efficient computability. We seek to improve this suboptimal practice in the context of physics simulations, where backpropagating feedback through many unrolled time steps is considered crucial to acquiring temporally coherent behavior. The alternative vector field we propose follows from two principles: physics simulators, unlike neural networks, have a balanced gradient flow, and certain modifications to the backpropagation pass leave the positions of the original minima unchanged. As any modification of backpropagation decouples forward and backward pass, the rotation-free character of the gradient field is lost. Therefore, we discuss the negative implications of using such a rotational vector field for optimization and how to counteract them. Our final procedure is easily implementable via a sequence of gradient stopping and component-wise comparison operations, which do not negatively affect scalability. Our experiments on three control problems show that especially as we increase the complexity of each task, the unbalanced updates from the gradient can no longer provide the precise control signals necessary while our method still solves the tasks. Our code can be found at https://github.com/tum-pbs/StableBPTT.

## 1 INTRODUCTION

Simulators are gaining popularity as an alternative way to train neural networks in an otherwise data-dominated deep learning field. These computational replicas of real-world phenomena can be directly built into the training loop and, if differentiable, combined with gradient-based optimizers. This was demonstrated in numerous areas, from fluid dynamics to robotics (Amos & Kolter, 2017; Toussaint et al., 2018; Schenck & Fox, 2018; Ingraham et al., 2018). The close interaction between neural network and simulator provides faster training than sample-inefficient trial-and-error search, does not suffer from the presence of multimodal solution modes, and integrates outlier cases into training as they occur.

In modern training setups, these simulator-network constructs are typically unrolled over many time steps, enabling the network to overlook long time spans and acquire a sense of long-term plausible behavior—that is, long-range planning instead of greedy solutions in control tasks (Belbute-Peres et al., 2018; Holl et al., 2020), temporal coherence instead of abrupt jumps in surrogate models (Kim et al., 2019; Stachenfeld et al., 2021), or long-term stability instead of diverging trajectories in error correction tasks (Um et al., 2020; Kochkov et al., 2021). As the time step is typically short to ensure numerically accurate results, the number of these time steps has then to be large to cover a sufficient time span, leading to a sequence of many recurrent network and simulator calls in each training step.

From a mathematical point of view, optimizing such long unrollments is a difficult problem: recurrently evaluating the same operator quickly creates an unbalanced optimization landscape. This is similar to how repeatedly multiplying a variable with itself leads to a high-order polynomial $f(x) = x^n$ with an increasingly impractical gradient: the larger $n$, the more $f$ and its derivatives vary in size. This is a simple form of what in the context of neural networks is often called the "exploding and vanishing gradient problem", a situation in which gradient-based optimizers perform

poorly (Goodfellow et al., 2016). The gradient flow in such unrollments has been widely studied under the name "Backpropagation Through Time" (Werbos, 1988) and addressing its mathematical weaknesses will facilitate the effective learning of temporal relationships.

Complementary to the existing approaches that focus on model architectures or loss formulations, we target the backpropagation step. As outlined above, the gradient field is only one among many vector fields that can guide a neural network toward an optimal configuration. We employ gradient stopping, one of the simplest ways to use that freedom, to propagate feedback only through the physics simulator but still over the full time trajectory. This has a positive effect on the magnitude of the update because in physical systems, the rate of change over time is typically limited, for instance by conservation laws. As any gradient stop not only stops feedback propagation but in general also literally stops the resulting vector field from being a rotation-free gradient field, the flow lines of this new vector field do not have to be perpendicular to minima surfaces anymore. This causes new challenges for existing optimizers, for instance, momentum terms can clash with spiraling flow lines around a minima and prevent convergence. Since the regular gradient field gives the correct direction near minima, our final algorithm performs only updates in those components that coincide in sign with the original gradient to counteract rotation. Both aspects above are central contributions of our work: the specific form of employing gradient stopping and the compensation of rotations are, to the best of our knowledge, novel. We evaluate the improvements of this algorithm on three control tasks: a guidance-by-repulsion model, a cart pole scenario, and a quantum system. Interestingly, the proposed algorithm fares especially well once the complexity of these scenarios is increased.

## 2  PROBLEM FORMULATION

We consider training setups in which a neural network $N$ parametrized by $\theta$ and a simulator $S$ interact with each other over several time steps $n$. Such setups are typically of the form:

$$
\begin{aligned}
c_i &= N(x_i, \theta) \ \text{ for } i = 0, 1, ..., n \\
x_{i+1} &= S(x_i, c_i) \ \text{ for } i = 0, 1, ..., n \\
L &= \frac{1}{2}(x_n - y)^2
\end{aligned}
\tag{1}
$$

A physical context for this setup would be a control task, where the goal is to transform an initial state $x_0$ into a target state $y$ via a controllable external force $c$. To induce that transition, a neural network $N$ estimates the control value $c_i$ for each time step $i$ based on the current physical state $x_i$. The effect of this action is then computed via the simulator $S$, which outputs the next physical state $x_{i+1}$. After repeating this process for a specified number of times $n$, a time trajectory is constructed with a final state $x_n$, whose distance $L$ to the target state $y$ is measured with a quadratic loss function. Besides control tasks, training setup 1 has also been applied to improve temporal coherence in error correction, reduced order modeling, and surrogate tasks (Belbute-Peres et al., 2018; Bar-Sinai et al., 2019; Wang et al., 2020a). We also introduce notations for the time stepping operator $T$, mapping $x_i$ to $x_{i+1}$, and its $n$-times application $F$, mapping the initial state $x_0$ to the final state $x_n$:

$$
\begin{aligned}
T(x_i, \theta) &= S(x_i, N(x_i, \theta)) = x_{i+1} \\
F(x_0, \theta) &= \underbrace{T(T(...(T(x_0, \theta), \theta)...), \theta)}_{\text{n-times}} = x_n
\end{aligned}
\tag{2}
$$

Finding network parameters $\theta$ that minimize the loss $L$ by using the gradient is inherently difficult: on the level of linear approximations, recurrently applying the same operator $T$ corresponds to a repeated multiplication by the same Jacobian matrix. As a result, the output of the backpropagation pass, the gradient, is dominated by the self-enforcing and self-damping mechanics of eigenvectors belonging to large and small eigenvalues. This means the gradient is no longer representative for the deviation between $x_n$ and $y$ and therefore a questionable choice for optimization.

**Illustration with a toy example:**  We illustrate these difficulties of the gradient for a simple case of 1 that will serve as the guiding example throughout our paper. We choose:

$$
\begin{aligned}
N(x_i, \theta) &= \theta_1 x_i^2 + \theta_2 x_i \\
S(x_i, c_i) &= x_i + c_i
\end{aligned}
\tag{3}
$$

While the choice of the simulator $S$, an identity map in the physical state $x_i$ plus control $c_i$, may appear overly simple at first, it allows us to avoid unrelated problems, such as discretizing differential operations or processing different physical time scales, and instead focus on the mathematical

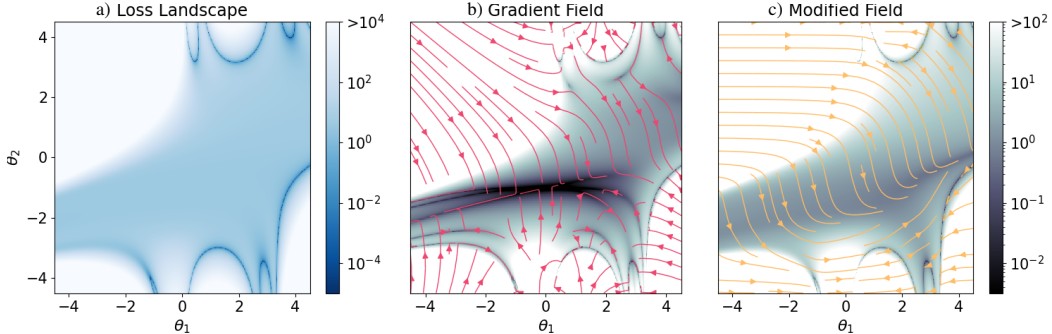

Figure 1: Toy example: a) loss landscape, b) regular gradient field (flow lines in red), c) modified vector field (yellow). For b) and c) the background color shows the L2 norm of the vectors. To improve optimization we trade an unbalanced gradient field for a balanced but rotating vector field.

essence of Backpropagation Through Time. In addition, choosing a polynomial controller provides non-trivial solutions yet keeps the mathematical expressions simple: composition of polynomials is intuitive while the composition of other analytical functions, e.g. tanh, quickly gives convoluted expressions. Most importantly, we restrict ourselves to two optimization variables $\theta_1$ and $\theta_2$. This enables us to visualize the optimization landscape in its entirety and convey a sense of what the dynamics of an optimization algorithm look like. In Figure 1, we depicted this toy example for initial state $x_0 = -0.3$, target $y = 2.0$ and $n = 4$ time steps.

Figure 1a and b show loss landscape and gradient field. The unbalanced landscape consists of an exploding outer area (white) with large gradients, a flat area mainly in the middle (light blue) and minima (dark blue). The imbalance worsens with increasing $n$, visualizations can be found in A, as the exponential effects at play increase the order of the saddle point, hence the vanishing gradient area extends and the minima become sharper. Gradient-based optimization works best when the gradient size is a measure of optimality, i.e. large gradients far away from minima, and small gradients nearby. Here, such a relationship is not only disrupted but inverted: with increasing $n$, ever more sharper minima with ever larger gradients around them, and farther away a growing area of near-constant loss with vanishing gradients. In the worst case, an optimizer would fight tediously through the area of vanishing gradients just to shoot off once a minimum comes within reach. Under these circumstances, we cannot expect gradient optimizers to do well. In the appendix C, we further study these landscapes for more complex systems such as a linear quadratic regulator or the trained networks in our later experiments.

## 3 METHOD

To improve this situation, our method creates a more suitable vector field for optimization:

### 3.1 MODIFYING BACKPROPAGATION

The backpropagation pass is where the gradient imbalance originates; modifying this step is therefore the natural starting point. In our equations, we use $d$ for the total derivative, $d_\theta L$ for the gradient of the loss, and $d\!\!\!/_\theta L$ for the corresponding result of the modified backpropagation. To begin with, we restrict ourselves to modifications that only change $d_\theta F$:

$$L = \frac{1}{2}\big(F(x_0, \theta) - y\big)^2$$
$$d_\theta L = \big(F(x_0, \theta) - y\big) \cdot d_\theta F \tag{4}$$
$$d\!\!\!/_\theta L = \big(F(x_0, \theta) - y\big) \cdot d\!\!\!/_\theta F$$

This restriction is motivated by considering critical points of the non-modified, regular gradient, i.e. points with $d_\theta L = 0$. We distinguish these points depending on whether or not $F(x_0, \theta) = y$. If yes, such a critical point is a global minimum and the modified update $d\!\!\!/_\theta L$ will also vanish, i.e. criticality of global minima is conserved. If not, such a critical point is not a global minimum, for instance, a non-global local minimum or a saddle point. For these, $d_\theta F$ is 0 while $d\!\!\!/_\theta F$ needs not to be and neither needs $d\!\!\!/_\theta L$. Therefore, these points are removed through such a modification.

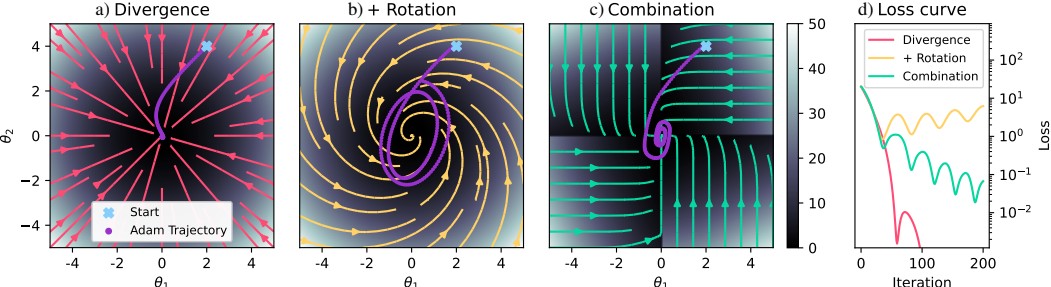

Figure 2: Minimization of a loss function with Adam using a) the gradient field b) a rotating vector field c) our combined vector field constructed from the vector fields in a) and b). Background color indicates vector length. d) shows the loss curves. The rotational contribution in b) prevents Adam from converging while our combined vector field in c) allows Adam to approach the minimum.

From the many vector fields that could be constructed in this way, we have to choose one that addresses the issues of vastly varying gradients across the optimization landscape. We decide to block any feedback flowing over the network $N$ to its input $x$ at any time step. To implement this, we use gradient stop operations that set $\not{\partial}_x N = 0$, while all other partial derivatives are the same as in the unmodified case. We refer to $\not{d}_\theta L$ computed with these modifications during backpropagation as *modified* update in the following. Such a partial stop of the feedback backpropagation does not violate our original goal to carry feedback back over the full temporal trajectory since the physics path $\partial_x S(x, c)$ remains untouched. Backpropagating only along this path improves the situation of the gradient sizes as most physics simulators come with a well-behaved gradient flow; the key lies in separately considering the partial derivatives for $x$ and $c$. While it is true that a physical optimization task to find a control $c$ can be arbitrarily ill-conditioned, we do not repeatedly propagate through $\partial_c S(x, c)$ but only rerun through $\partial_x S(x, c)$. A physical state $x$ and its successor hold a close relationship to each other reflecting the physical characteristics of the system. Norms computed on them can correspond to various quantities that are either conserved, such as a probability sum in quantum systems, or at least controlled, such as energy in a mechanical system when a state is brought from one energy level to another. Such restrictions on the changes of a state $x$ reflect themselves also in limitations, upper and lower, on the size of its gradient $\partial_x S(x, c)$.

In our toy example, the simulator $S$ is an identity map in the physical state $x$, therefore the gradient flow over the physics path is trivially optimal. We construct our modified vector field by neglecting any feedback from the polynomial controller $N$ to the state $x$ and show this alternative vector field in Figure 1c. We observe improved optimization landscape: the saddle point and its extended area of vanishing gradients vanish and the new gradient field also visually resembles the loss landscape more closely, indicating improved conditions for optimization with gradient-based optimizers.

## 3.2 Emergence of Rotational Vector Fields

The observed improvements come at a cost: any modifications of the backpropagation pass decouple it from the corresponding forward pass and can destroy the rotation-free character of the gradient field, an unfamiliar situation in optimization. In general, rotation of a vector field can be determined from the antisymmetrical part of its Jacobian, requiring us to further differentiate equations 4.

$$
\begin{aligned}
d_\theta^2 L &= \left(F(x_0, \theta) - y\right) \cdot d_\theta^2 F + (d_\theta F)^2 \\
d_\theta \not{d}_\theta L &= \left(F(x_0, \theta) - y\right) \cdot d_\theta \not{d}_\theta F + d_\theta F \cdot \not{d}_\theta F
\end{aligned}
\tag{5}
$$

For unmodified backpropagation, $d_\theta^2 L$ is a Hessian matrix, therefore symmetric and free of rotation. If modifications are applied, such an argument cannot be made. Even near global minima, where the second summands dominate, the inequality of $d_\theta F$ and $\not{d}_\theta F$ makes $d_\theta \not{d}_\theta L$ non-symmetric, creating rotation in our modified vector field $\not{d}_\theta L$.

This can be directly seen in our toy example. The gradient of a function points in the direction of steepest ascent, making its flow lines orthogonal to level sets and curves of minima, clearly fulfilled for the regular gradient $d_\theta L$ in Figure 1b. However, it further means that curves of minima close to each other are surrounded by abruptly changing gradients, another aspect worsening with more time

steps and visible in figure 6. In contrast in Figure 1c, the rotation in our modified field $\not{d}_\theta L$ allows flow lines to intersect minima curves non-orthogonally. This additional freedom balances the vector field, removing abrupt variations near close minima but also around the original saddle point.

So far we have explained why $\not{d}_\theta L$ is rotating, but it is not clear yet why this could be a problem. In Figure 1c, rotation adds a tangential movement along minima curves that should at worst lead to a slower convergence towards a minimum. We consider now the case when rotation adds a circular movement around minima as the second way of how a rotational vector field can be organized around a minima manifold. For a simple illustration, we consider a quadratic loss $L$ around a minimum at $(0,0)$, a rotation-free vector field $D$, a purely rotating vector field $R$, and try to minimize the loss with Adam (Kingma & Ba, 2014) and different vector fields:

$$
\begin{aligned}
L(\theta_1, \theta_2) &= \theta_1^2 + \theta_2^2 \\
D(\theta_1, \theta_2) &= \partial_\theta L = (-\theta_1, -\theta_2) \\
R(\theta_1, \theta_2) &= (-\theta_2, \theta_1)
\end{aligned}
\tag{6}
$$

Figure 2a shows the behavior of Adam moving in $D$, the simple case of a gradient field where Adam succeeds easily. Next, 2b shows Adam moving the vector field $D + 2R$, a vector field with a rotational contribution, corresponding to what we encounter when modifying backpropagation. The circular contribution makes the flow lines spiral around the minimum and instead of following those, Adam constantly overshoots, ending up in a circle-like trajectory around the center without converging. This is caused by the momentum terms in the update procedure of Adam but also partly by the discrete nature of the algorithm. For completeness, Figure 2d contains the corresponding loss curves. Together with the other, tangential contribution, this completes our picture of the positive and negative effects that rotation can cause in an optimization landscape.

### 3.3 COMBINATION TO COUNTERACT ROTATION

To address the problem Adam had in the rotating vector field, it is helpful to understand why optimization in a rotational field is to some degree uncharted territory. Discrete time-stepping methods used in numerical simulation just like optimizers perform iterative updates to follow the flow lines of a vector field. After all, gradient descent steps are nothing different than Euler steps in the negative gradient field. In numerical simulation, there exists an abundance of methods designed to closely follow the flow of a vector field, no matter if rotational or not. However, these are not applicable here since in optimization, it is important to accelerate along flow lines in flat valley environments, or even leave the flow lines to avoid suboptimal local minima. Since these are contradicting goals, a possible solution cannot be obtained by changing the way the modified update $\not{d}_\theta L$ is processed. An example is momentum, which is widely accepted to give the explorative behavior required in optimization but it is exactly what prevented convergence to the minima in Figure 2b.

Therefore, we instead focus on jointly processing $d_\theta L$ and $\not{d}_\theta L$. Although we criticized many properties of the regular gradient $\partial_\theta L$ in Figure 1a, its direction near global minima is one thing that is undoubtedly good. This motivates that for our final update vector $u$, we introduce a component-wise condition that sets the $j$-th component to zero if $d_{\theta_j} L$ and $\not{d}_{\theta_j} L$ have opposite signs:

$$
u_j = \begin{cases} \not{d}_{\theta_j} L & \text{if } \operatorname{sign}(d_{\theta_j} L) = \operatorname{sign}(\not{d}_{\theta_j} L) \\ 0 & \text{otherwise} \end{cases}
\tag{7}
$$

We call this quantity $u$ the *combined* update. Geometrically, $u$ is at most orthogonal to $d_\theta L$ and to $\not{d}_\theta L$. In our optimization context, it has several desirable properties: it has the same well-balanced behavior as $\not{d}_\theta L$ but improved convergence near rotational minima and compatibility with momentum, as can be seen in Figure 2c.

### 3.4 COMPUTATIONAL COMPLEXITY

Computing the modified update $\not{d}_{\theta_j} L$ requires fewer operations as we no longer have to backpropagate through any path including a network output to network input connection. Computing the combined update $u$ requires two backpropagation passes and a component-wise sign comparison that is negligible in comparison to backpropation. Therefore, a straight-forward sequential implementation of the two backpropagation passes leads to doubled runtime and memory requirements. However, since they are independent, a parallel implementation is possible to avoid the runtime increase. On top of that, the forward pass and its recorded intermediate results required for backpropagation are the same. Hence, most of the memory increase can be avoided as well. In practice, the additional computational requirements are manageable, as we show in appendix B.1.

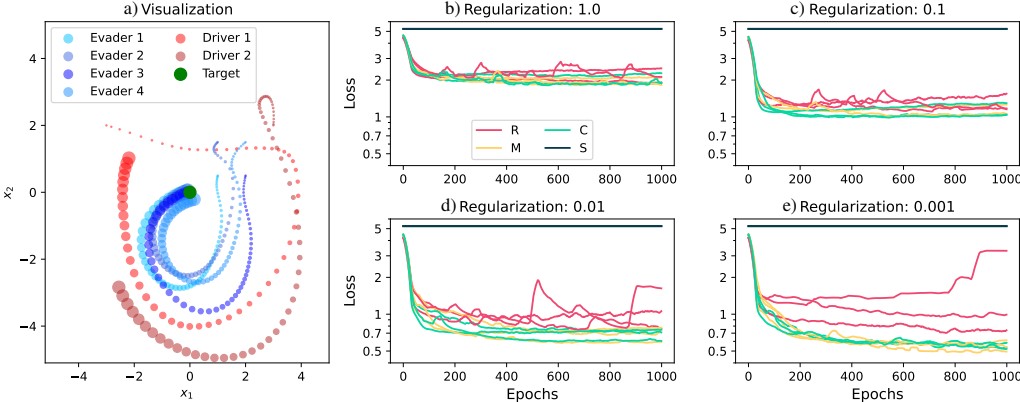

Figure 3: Guidance-by-repulsion model: a) visualization, smaller circles indicate the configuration at earlier points in time, b) - e) learning curves for different regularization coefficients. *(S)* is not able to learn. With less regularization, *(C)* and *(M)* increasingly outperform *(R)*. Same-color curves differ by clipping mode. Curves were smoothed over 20 epochs for clarity.

## 4 EXPERIMENTS

In our experiments, we compare our proposed combined update vector *(C)* to the established regular gradient *(R)*. Additionally, we include the modified update *(M)* that only backpropagates through the physics simulator as an ablation study. As a further baseline, we consider a stopped update vector *(S)* resulting from backpropagating only through the last unrollment step. This is used in some works (Prantl et al., 2022; Brandstetter et al., 2022) to trade temporal look-ahead for stability, but can be expected to perform less well due to the truncated temporal feedback. These four techniques are comparable since the forward pass, and hence the loss formulation is the same for all. Based on our derivation, we investigate the following hypotheses in our experiments:

  (i) *(M)* and *(C)* will outperform *(R)*. (Motivated by their more balanced updates.)

 (ii) *(C)* will outperform *(M)*. (Thanks to its ability to counteract rotation.)

(iii) These differences become more apparent as the task complexity increases. (In easy tasks the negative effects of unbalanced updates can be more easily corrected.)

To investigate these hypotheses, we study three control tasks: a guidance-by-repulsion model, a cart pole swing-up, and a quantum control problem. We also present a cart pole variation with walls as a simple contact task in the appendix D The differential equations describing these systems and the simulation details can be found in appendix B. Across the tasks, we vary model architecture and choose either a final loss, where the physical state is compared to the target state only at the last time step, or an accumulated loss, where the physical state is compared to the target state after every time step. We choose Adam as optimizer to process the four different backpropagation vectors with a learning rate of 0.001 and a batch size of 8. This set of hyperparameters performed well across our tests, but we include an extensive search over 792 training runs with variations of the optimizer, learning rate and batch size in the appendix.

All following learning curves show end-of-epoch losses on test data against the number of epochs; we provide runtime data in appendix B.1. The plots contain curves of four different colors for the four different backpropagation methods, the regular update *(R)* in red, modified *(M)* in yellow, combined *(C)* in green, and stopped *(S)* in black, and for each method, three same-color curves corresponding to three clipping modes (value, norm, or no clipping). Including different clipping techniques serves a double purpose: showing problems from exploding gradients are not resolved by simple cut-offs and highlighting how consistently the methods behave over different runs.

### 4.1 GUIDANCE-BY-REPULSION

This coupled system of differential equations from the class of collective behavior models (Ko & Zuazua, 2020) has two types of agents, controllable *drivers* and non-controllable *evaders*. Several

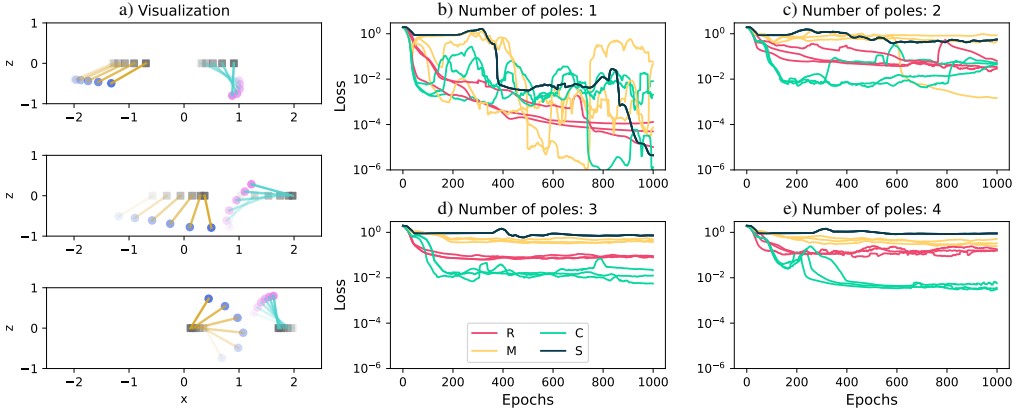

Figure 4: Cart pole: a) visualization of the two-poles task, upper plots and more transparent style indicate earlier points in time, for easier visualization the two poles are plotted attached to two carts instead of one, b) - e) learning curves for 1-4 poles. With more poles, *(C)* outperforms *(R)*, *(M)* and *(S)*. Same-color curves differ by clipping mode. Curves were smoothed over 20 epochs for clarity.

types of interactions are present such as a flocking term for evaders, a spreading term for drivers, and a repelling term causing evaders to run away from nearby drivers. As the last term grows quickly when drivers come near evaders, this resembles the interactions found in contact systems In conjunction, they resemble the dynamics of a group of shepherds, see Figure 3a for an illustration. The learning goal is to find a control strategy for two drivers that will move four evaders to a specified position. We increase complexity and allow for more sophisticated behavior by decreasing a regularization coefficient, which allows drivers to go faster and change direction more frequently. We unroll over 60 time steps and use an accumulated loss. The controller is implemented as a fully connected network with 13004 parameters. We train with 256 initial states, created by placing a group of four evaders randomly at a minimum distance around the target state and the two drivers farther away in the outer parts of the system.

Figure 3b-e shows the results of this control task for different regularization values. In all cases, *(S)* does not learn a meaningful control strategy with a constant loss of around 5. With the strongest regularization, the other three techniques are behaving similarly with a loss ranging from 2 to 3. Only *(R)* shows a slightly more unstable behavior with slightly higher loss values. As we decrease regularization, *(R)* falls back, and *(M)* shows a similar performance as *(C)*. This supports hypothesis (i) and (iii) but not (ii).

## 4.2 CART POLE SWING-UP

The widely-used cart pole system consists of a pendulum attached to a controllable cart. The goal is to move the cart in such a way that the pendulum exits its lower position and swings up to reach its highest possible point. In our variant, we have several poles to increase the difficulty of the task, see Figure 4a for an illustration. When the poles' starting positions differ, the network is required to apply a stronger and faster varying control to swing up all of the poles. In our setup, we unroll the cart pole simulator over 100 time steps and apply a loss on the final physics state. We use a fully connected network architecture with around 11000 trainable parameters. We train with 256 different initial configurations, where the poles are randomly swung out by an angle up to $30°$.

Figure 4b-e shows the result of these training runs, where the number of poles is increased from 1 to 4 to vary complexity. In the easiest 1-pole task, all methods are able to solve the task with final loss values of around 0.01 and lower. This picture changes in favor of our combined update *(C)* for the harder tasks: while there are some outlier cases in the 2-poles case, *(C)* fares consistently better than the *(R)*, *(M)*, and *(S)* with 3 poles and more, and is the only method able to solve the task with a loss below 0.01. This is in line with hypothesis (iii), and (ii) for 2 to 4 poles; (i) is partially fulfilled, *(R)* is worse then *(C)* but better than *(N)* is these training runs.

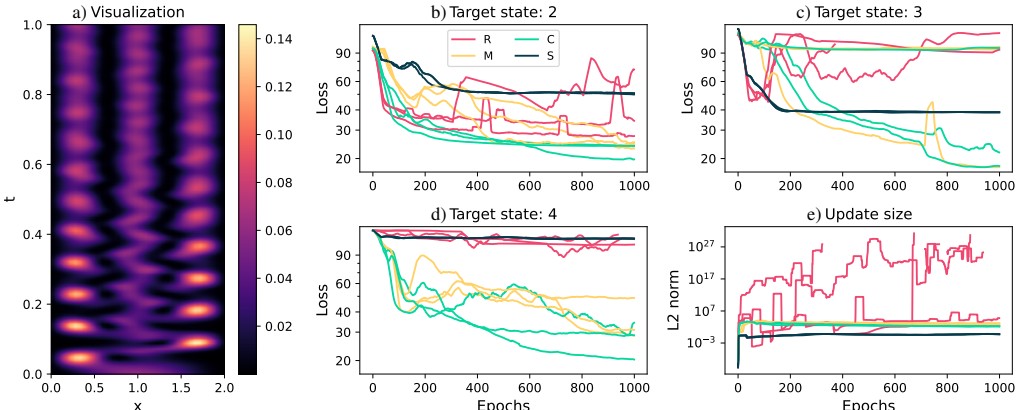

Figure 5: Quantum control: a) visualization of a state transition over time $t$ and space $x$, background color indicates the probability density , b) - d) learning curves for different target states , e) update size. For higher target states, *(C)* outperforms *(R)*. Same-color curves differ by clipping mode. Curves were smoothed over 20 epochs for clarity.

## 4.3 QUANTUM CONTROL

In our last experiment, we target the quantum dipole task (Von Neumann, 2018), a control problem formulated on the Schroedinger equation. The goal is to find a control signal to induce transitions from lower to higher energy states, see 5a for a visualization of such a transition. The higher the energy difference, the more difficult the task. Concretely, we build the data set for the initial state from superpositions of the ground and first excited state and consider three tasks with the target state being either the second, third or fourth excited state. We consider the case of a accumulated loss, unroll for 128 time steps and use a convolutional neural network with 11701 parameters.

Figure 5b-d shows the learning curves. Reaching the second state is the simplest task, for which the different methods perform the most similarly. Nonetheless, it is already apparent that *(M)* and *(C)* outperform *(R)* and *(S)*. For the third state, *(S)* still achieves some improvement but *(R)* is no longer able to solve the task. Best performing are one run of *(M)* and two runs of *(C)* minimizing the loss to 30 and below. For the forth state, the hardest case, the picture becomes even clearer: while *(R)* and *(S)* do not make any progress, *(M)* is able to achieve a decent result, and *(C)* performs even better with a run achieving a loss of 20. This again is in line with (i) and (iii), and weakly supports (ii) with individual instances of good performance for *(M)*, but *(C)* performing better in general.

This example encapsulates several key arguments of our derivation: in mathematical formulations of quantum systems, the squared physical state is interpreted as probability density, and its sum has to add up to one. Neither natural time evolution nor external forces can violate this property, making the L2 norm of the state a perfectly conserved quantity. This forces the eigenvalues of the simulator to be one and therefore feedback can be backpropagated without creating an imbalance. To visualize this, we tracked the L2 norms of the backpropagation results for the last, most difficult task in figure 5e. The graph shows that *(R)* varies by more than 35 orders of magnitude while the magnitude of the *(M)* and *(C)* is preserved over the entire training. This illustrates our reasoning and gives an intuition for the improved convergence of networks trained with the modified and combined updates.

## 4.4 HYPERPARAMETER STUDY

We conclude with a short summary of our hyperparameter study (details can be found in the appendix). For the cart pole task, we observed that 24 *(M)*-runs versus only 7 *(R)*-runs minimized the loss successfully to 0.01, supporting hypothesis (i). The guidance-by-repulsion system behaves similarly. Noteworthy for quantum control, not a single training run with *(R)* solved the task. Regarding *(M)* versus *(C)*, both worked equally well on two tasks while on the cart pole, *(C)* performs better, adding up to a support of hypothesis (ii). In summary, our experiments provide strong evidence in favor of hypothesis (i) and (iii), and evidence for hypothesis (ii). This constitutes, to the best of our knowledge, a fair and comprehensive analysis of our method.

## 5 RELATED WORK

Machine learning techniques have a rich history of integrating physical models (Crutchfield & McNamara, 1987; Kevrekidis et al., 2003; Brunton et al., 2016). For instance, researchers have employed machine learning to discover formulations of partial differential equations (Long et al., 2017; Raissi & Karniadakis, 2018; Sirignano & Spiliopoulos, 2018), explore connections to dynamical systems (Weinan, 2017), and leverage conservation laws (Greydanus et al., 2019; Cranmer et al., 2020). Previous investigations have also uncovered discontinuities in finite-difference solutions through deep learning (Ray & Hesthaven, 2018) and have concentrated on enhancing the iterative performance of linear solvers (Hsieh et al., 2019). Another interesting avenue for the application of deep learning algorithms lies in Koopman operators (Morton et al., 2018; Li et al., 2019). Differentiable simulations (Thuerey et al., 2021) have been combined with machine learning to describe a wide range of physical phenomena including molecular dynamics (Wang et al., 2020b), weather (Seo et al., 2020), rigid-bodies (Battaglia et al., 2013; Watters et al., 2017; Bapst et al., 2019), fluid flows (List et al., 2022), or robotics (Toussaint et al., 2018). Gradient-based optimization can be used in conjunction to solve inverse problems (Liang et al., 2019; Schenck & Fox, 2018). By now, several frameworks exist for differentiable programming (Schoenholz & Cubuk, 2019; Hu et al., 2020; Holl et al., 2020). Unrolling can be regarded as an advanced technique in the long history of fusing machine learning with scientific computing to improve learning of temporal relationships.

Backpropagation Through Time (BPTT) leads to the exploding and vanishing gradient problem (Bengio et al., 1994) and few techniques are known to address this problem in unrollments including a simulator. Specialized architectures, such as long short-term memory (Hochreiter & Schmidhuber, 1997), avoid exploding gradients but are targeted to work on a priori known and therefore normalizable data sets, whereas in control tasks the scales of the network outputs are beforehand unknown. One applicable strategy with simulators is to initialize networks to output zero and start exploring the physical system from its natural, control-free time evolution outside an exploding gradient region (Degrave et al., 2019). In terms of loss formulation, reducing the unrollment steps limits the extent of the gradient imbalance at the cost of encouraging greedier actions on a shorter time horizon (Grzeszczuk et al., 1998). Truncating BPTT has a long tradition to stabilize long recurrent rollouts (Sutskever, 2013). Recently, it has been proposed for controlling and predicting different physical systems (Xu et al., 2021; Brandstetter et al., 2022; Prantl et al., 2022). These approaches for truncation primarily reduce unrollment steps, alleviating the underlying issues without directly addressing them. Apart from backpropagation, suggested improvements include clipping (Sutskever, 2013) or the use of higher-order optimization (Martens & Sutskever, 2011), which reverts any observed gradient imbalance in a computationally expensive process, limiting practical applicability. We presented a scalable method with a similar effect by analyzing the structure of the computation graph. In particular in Reinforcement Learning, many tasks include contact (Freeman et al., 2021), destabilizing gradient methods (Metz et al., 2021; Suh et al., 2022; Antonova et al., 2023) , and making the use of specialized techniques necessary (Parmas et al., 2018; Qiao et al., 2021).

## 6 CONCLUSION

We focused on improving the training of recurrent simulator-network setups, an important topic that facilitates the learning of more complex temporal relationships. For this, we targeted the backpropagation pass and constructed an alternative vector field for optimization without the widely known weaknesses of regular gradients. The modifications in the form of gradient stopping and comparison operations were motivated by the well-behaving gradient flows of physics simulators and the emerging rotation in this new vector field, respectively. We demonstrated our method successfully on three control problems with a diverse physical nature.

A current limitation of our work is that we have not yet evaluated contact-rich scenarios, a potentially very interesting avenue for our method. Their discontinuous dynamics have the potential to conflict with our assumption that physical simulators have a well-behaved gradient flow. An interesting direction for future work will also be to re-evaluate various limitations associated with temporal unrollments: an improved optimization method could reveal that the issue was not the formulation of the problem, but rather the way it was solved. Since the actual modifications we proposed are simple, our work could provide the starting point for research into more complex modifications of the backpropagation pass. We believe this to be a promising area of research because, despite our investigations of two of them above, all the other vector fields surrounding the minima of recurrent learning setups are still unexplored.

REPRODUCIBILITY STATEMENT

To ensure reproducibility of our work we will publish the source code for our experiments upon acceptance. Details on our experiments are also provided in the appendix.

ACKOWLEDGEMENTS

This work was supported by the ERC Consolidator Grant CoG-2019-863850 SpaTe, and by the DFG SFB-Transregio 109 DGD. We would also like to express our gratitude to the reviewers and the area chair for their helpful feedback.

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

# Appendix

## A  TOY EXAMPLE: VARYING THE NUMBER OF TIME STEPS

In Figure 6, we illustrate how the landscape and gradients in the toy example 3 change depending on the number of time steps $n$. We show $n = 2$, 4, 6, 8 and 10 steps. The optimization difficulties with increasing $n$ are sharper minima (left column) that are harder to approach, and more minima curves closer to each other (middle column) with highly varying gradients in between. A saddle point increasing in order with an extending area of vanishing gradients is visible as a dark area in the middle column, and exploding gradients in the outer areas (also middle column). As one can see, these quantities with highly varying magnitudes are not only a problem for optimization but also for visualization; the flow lines of the gradient can barely be plotted (middle column, bottom row). In contrast, our modified update has more balanced update sizes (right column).

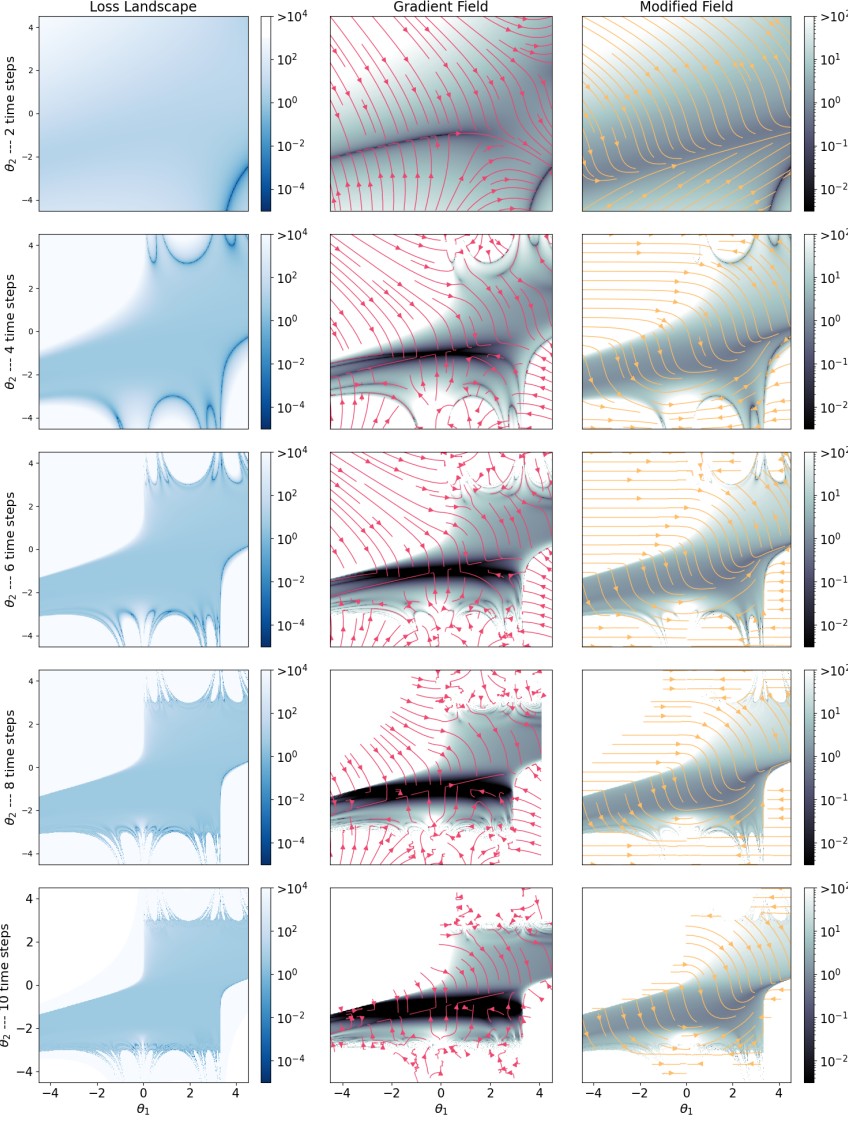

Figure 6: Further visualization of the toy example for 2, 4, 6, 8 and 10 time steps from top to bottom. Loss landscape on the left, regular gradient field in the middle, modified vector field on the right. With time steps increasing, optimization becomes for difficult due to sharper minima and a higher-order saddle point.

## B  EXPERIMENTAL DETAILS

Here we provide details on our three control tasks: guidance-by-repulsion, cart pole swing-up and quantum control. Moreover, we present run time data and a hyperparameter study for the four methods considered in the experiments, the regular gradient *(R)*, the modified update *(M)*, the combined update *(C)* and the stopped update *(S)*. Below, these methods are visualized with the same colors as before: red for *(R)*, dark green for *(S)*, and the modified version *(M)* in yellow, while the combined update *(C)* is colored green.

### B.1  RUNTIME COMPARISON

To illustrate the computational effort of each method, Table 1 lists the epoch durations for the four considered backpropagation methods. As expected, *(S)* is the fastest. For guidance-by-repulsion and cart pole swing-up, *(M)* is just as fast as *(R)*, and *(C)* needs almost double the time. In contrast for quantum control, *(M)* is even faster than *(R)* and *(C)* is only slightly slower than *(R)*. Especially the quantum control case demonstrates that *(M)* can be computed faster and *(C)* almost as fast as *(R)*, which is in line with our theoretical argumentation in 3.4.

Table 1: Epoch duration in seconds for our three control tasks

| Method | (R) | (M) | (C) | (S) |
|---|---|---|---|---|
| Guidance-by-Repulsion | 3.40 | 3.42 | 6.50 | 1.54 |
| Cart Pole Swing-Up | 1.57 | 1.58 | 3.03 | 1.18 |
| Quantum Control | 11.97 | 8.42 | 12.34 | 7.10 |

### B.2  GUIDANCE-BY-REPULSION

**Differential equation and numerical simulation**    This coupled system (Ko & Zuazua, 2020) consists of drivers ($d$) and evaders ($e$) and follows a Newtonian dynamic, $i$ denoting the $i$-th driver, $j$ the $j$-th driver, $x$ their positions in a two-dimensional plane, dots time derivatives, and $F$ the sum of all forces acting upon them:

$$
\begin{aligned}
\ddot{x}_i^d(t) &= F_i^d \\
\ddot{x}_i^e(t) &= F_i^e
\end{aligned}
\tag{8}
$$

The total force on drivers is a sum of a friction force, a spreading force, and a control force. Each driver can be steered with two controllers $c_i^{(1)}$ and $c_i^{(2)}$:

$$
\begin{aligned}
F_i^d &= F_i^{fric-d} + F_i^{spread} + F_i^{con} \\
F_i^{fric-d} &= -\dot{x}_i^d \\
F_i^{spread} &= \frac{1}{5} \sum_{k \neq i} \frac{x_k^d - x_i^d}{||x_k^d - x_i^d||^2} \\
F_i^{con} &= c_i^{(1)}(x_i^d - B(x^d)) + c_i^{(2)}(x_i^d - P(x^d)) \\
B(x_d) &= \sum_k x_k^d \\
P(x_d) &= (-B(x^d)_2, B(x^d)_1)^T
\end{aligned}
\tag{9}
$$

$||\cdot||$ denotes the L2 norm. The total force on evaders is a sum of a friction force, a repulsive force, and a flocking force:

$$F_j^e = F_j^{fric-e} + F_j^{rep} + F_j^{flock}$$

$$F_j^{fric-e} = -4\dot{x}_i^e$$

$$F_j^{rep} = -15 \sum_k \frac{(x_k^d - x_j^e)}{||x_k^d - x_j^e||^2}$$

$$F_j^{flock} = \frac{1}{5} \sum_{l \neq j} g(||x_j^e - x_l^e||)(x_j^e - x_l^e)$$

$$g(p) = \frac{1}{2p^2} - \frac{1}{p}$$

(10)

We use $60$ Euler steps to simulate a time span of $T = 4$. For the training and test data set, we generate $256$ initial states with $2$ drivers randomly at a distance from $3$ to $4$ around the origin, and evaders randomly in a square of side length $1$ which is randomly placed at a distance from $1$ to $2$ around the origin. The goal is to guide the evaders towards the origin. The network controller is a fully connected neural network: input layer (input shape $(24)$), fully connected layer ($100$ features, $\tanh$-activation), fully connected layer ($100$ features, $\tanh$-activation), output layer (output shape $(4)$, no activation), all layers using weights and biases, totaling to $13004$ parameters. We use an accumulated loss function, computing the distance to the origin of each evader after every time step plus the L2 norm of the control values $c$ multiplied by a regularization coefficient. This coefficient is changeable to vary the complexity of the task. In the learning curves shown in the main part, we used Adam with a learning rate of $0.001$ and batch size of $8$ with three different clipping modes (value clipping to $1.0$, norm clipping to $1.0$, and no clipping) and trained for $1000$ epochs.

**Hyperparameter study** Figure 7 shows training runs for different learning rates and optimizers with batch size $8$. Figure 8 shows training runs for different learning rates and batch sizes with Adam. *(S)* is not able to solve this optimization task. Both *(M)* and *(C)* minimize the loss successfully and often behave similarly. In comparison, *(R)* sometimes achieves a just as good result but is more unstable with several instances of a much worse loss value, usually for higher learning rates. The quantitative summary in Table 2 confirms that *(M)* and *(C)* are both equally suited to solve this control task and clearly better than *(R)*.

Table 2: Guidance-by-repulsion: quantitative summary of our hyperparameter study

| Method | *(R)* | *(M)* | *(C)* | *(S)* |
|---|---|---|---|---|
| Loss of the best run | 0.590 | **0.465** | 0.495 | 5.24 |
| Average loss of the best 5 runs | 0.639 | **0.492** | 0.503 | 5.24 |
| Average loss of the best 25 runs | 0.839 | **0.547** | 0.560 | 5.24 |
| Number of runs below loss 0.5 | 0 | **2** | **2** | 0 |
| Number of runs below loss 0.8 | 11 | 37 | **38** | 0 |
| Total runs | 60 | 60 | 60 | 60 |

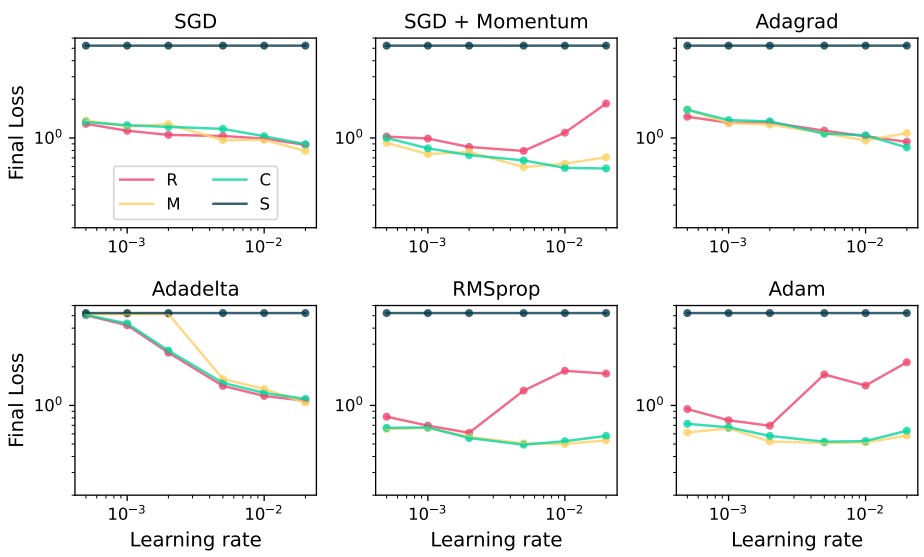

Figure 7: Guidance-by-repulsion, hyperparameter study with different learning rates and optimizers.

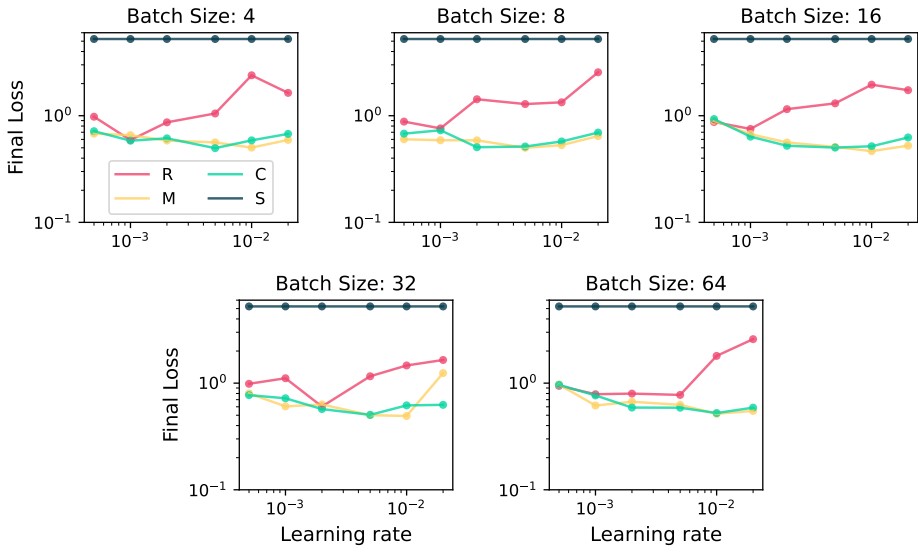

Figure 8: Guidance-by-repulsion, hyperparameter study with different learning rates and batch sizes.

## B.3 CART POLE SWING-UP

**Differential equation and numerical simulation** We consider $i$ poles attached to a cart and neglect friction. Our equations are a generalization of the classic cart pole equations with one pole (Florian, 2005). We use $x$ for the position of the cart, $\theta_i$ for the angle of pole $i$, and $F$ for the control variable. Furthermore, $A$,$B$, and $C$ are used to keep the notation compact. The remaining parameters are $g$,$m$, $M$, and $l$ are the gravity constant, masses of the components, and the length of the poles.

$$
\begin{aligned}
\ddot{x} &= \frac{F + mlC}{M} \\
C &= \sum_i \left( \dot{\theta}_i^2 \sin(\theta_i) - \ddot{\theta}_i \cos(\theta_i) \right) \\
\ddot{\theta}_i &= \frac{g \sin(\theta_i) - A_i \cos(\theta_i)}{B_i} \\
A_i &= \frac{F + ml\dot{\theta}_i^2 \sin(\theta_i)}{M} \\
B_i &= l \left( \frac{4}{3} - \frac{m \cos(\theta_i)^2}{M} \right)
\end{aligned}
\tag{11}
$$

For simulation, we set $g = 9.8$, $m = 0.1$, $M = 1.1$, $l = 0.5$ and vary the number of poles from $i = 1$ to $4$. We use $100$ semi-implicit Euler steps with a time step of $0.01$. For the training and test data set, we generate $256$ different initial configurations, where the poles are randomly swung out by an angle up to $30°$ and the goal is to swing all the poles up to the highest possible point. The network controller is a fully connected neural network: input layer (input shape $(2 + 2 \cdot i)$), fully connected layer ($100$ features, $\texttt{tanh}$-activation), fully connected layer ($100$ features, $\texttt{tanh}$-activation), output layer (output shape $(1)$, no activation), all layers using weights and biases, totaling to $10501 + 200 \cdot i$ parameters. We use a final loss function, computing the distance of the poles to the highest point only after the last time step. In the learning curves shown in the main part, we used Adam with a learning rate of $0.001$ and batch size of $8$ with three different clipping modes (value clipping to $1.0$, norm clipping to $1.0$, and no clipping) and trained for $1000$ epochs.

**Hyperparameter study** Figure 9 shows training runs for different learning rates and optimizers with batch size $8$. Figure 10 shows training runs for different learning rates and batch sizes with Adam. *(S)* is not able to solve this optimization task. *(R)* is better but rarely able to minimize the loss below $0.01$. *(M)* is second best, often better then *(R)* but worse than *(C)*. This can also be seen in the quantitative summary in Table 3, which shows *(M)* is an improvement over *(R)* and *(C)* is the best method in all metrics.

Table 3: Cart pole swing-up: quantitative summary of our hyperparameter study

| Method | (R) | (M) | (C) | (S) |
|---|---|---|---|---|
| Loss of the best run | 0.00194 | 0.00165 | **0.00145** | 0.644 |
| Average loss of the best 5 runs | 0.00273 | 0.00217 | **0.00161** | 0.803 |
| Average loss of the best 25 runs | 0.0694 | 0.00451 | **0.00386** | 0.915 |
| Number of runs below loss 0.002 | 1 | 2 | **6** | 0 |
| Number of runs below loss 0.01 | 7 | 24 | **37** | 0 |
| Total runs | 78 | 78 | 78 | 78 |

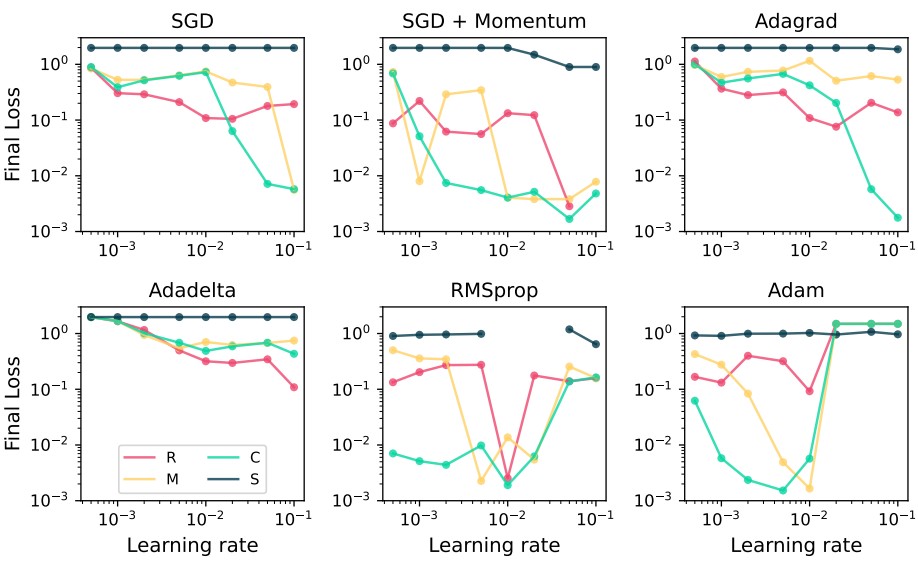

Figure 9: Cart pole swing-up, hyperparameter study with different learning rates and optimizers.

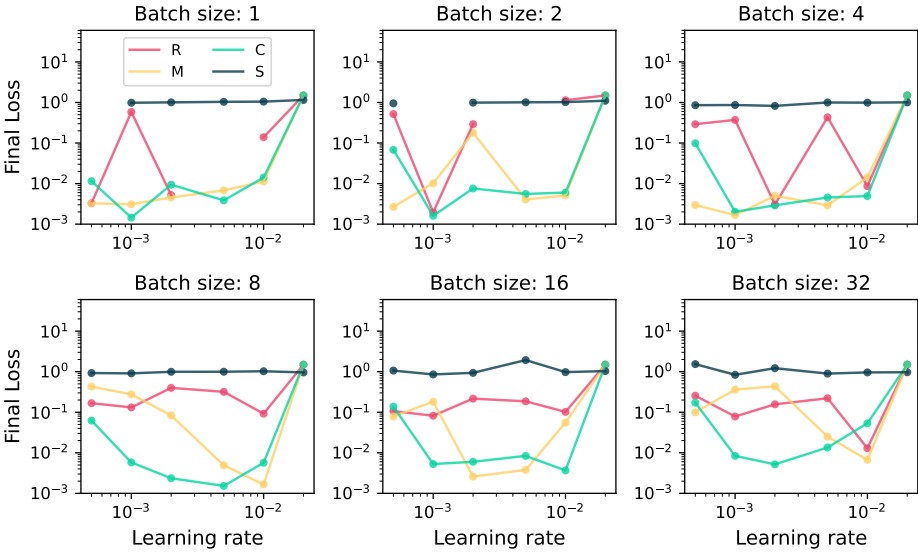

Figure 10: Cart pole swing-up, hyperparameter study with different learning rates and batch sizes.

### B.4 QUANTUM CONTROL

**Differential equation and numerical simulation:** The quantum dipole task is a temporal control problem on the Schroedinger equation (Von Neumann, 2018):

$$i\partial_t \Psi = \left( -\Delta + V + u(t) \cdot \hat{x} \right) \Psi \tag{12}$$

The notation is $i$ for the imaginary unit, $\partial_t$ for the time derivative, $\Psi$ for the quantum state or wave function, $\Delta$ for Laplacian, $V$ for an infinite-well potential, $u(t)$ the control function we are solving for, and $x$ for position operator. Similarity of quantum states is measured with an inner product loss:

$$L(\psi_1, \psi_2) = 1 - |\langle \psi_1, \psi_2 \rangle|^2 \tag{13}$$

We discretize the equation on a spatial domain $[0, 2]$ with 32 spatial discretization points. We simulate from physical time 0 to 1 in 128 time steps using a modified Crank-Nicolson scheme (Winckel et al., 2009). For training and test data set, we generate 256 random superpositions of ground and first excited state, both equally weighted, which serve as the initial states to the control problem. The target state is either the second, third or fourth excited state, depending on task difficulty. The network controller is a convolutional neural network: input layer (input shape $(32, 2)$), two-dimensional convolution layer (60 features, kernel $(3, 2)$, stride 2, $\texttt{tanh}$-activation), one-dimensional convolution layer(60 features, kernel 3, stride 2, $\texttt{tanh}$-activation), output layer (output shape $(1)$, no activation), all layers using weights and biases, totaling to 11701 parameters. We use an accumulated loss function, computing the similarity to the target state after each time step. In the learning curves shown in the main part, we used Adam with a learning rate of 0.001 and batch size of 8 with three different clipping modes (value clipping to 1.0, norm clipping to 1.0, and no clipping) and trained for 1000 epochs.

**Hyperparameter study** Figure 11 shows training runs for different learning rates and optimizers with batch size 8. Figure 12 shows training runs for different learning rates and batch sizes with Adam. *(R)* is not able to solve this optimization task. *(S)* generally fares similarly; only in individual cases it comes close to *(M)* and *C*, for instance when RMSprop was used as optimizer. Altogether it is clearly visible that *(M)* and *(C)* perform much better here, and generally exhibit a matching performance. This can also be seen in Table 4: the best performing runs achieved the best loss and counting the number of runs which solved the task well with a loss of 30 or below, we count 14 for both methods. With the inability of *(R)* to solve this problem and the very good performance of *(M)* and *(C)* across a wide range of hyperparameters, we consider this quantum control problem a true paragon of our advice on the beneficial gradient flow of physics simulators. The normalized states of the quantum system provide a well-behaved gradient flow that clearly benefits from the modified backpropagation pass.

Table 4: Quantum control: quantitative summary of our hyperparameter study

| Method | (R) | (M) | (C) | (S) |
|---|---|---|---|---|
| Loss of the best run | 85.6 | **19.0** | **19.0** | 38.4 |
| Average loss of the best 5 runs | 97.7 | **19.7** | 20.0 | 40.5 |
| Average loss of the best 25 runs | 99.4 | **28.9** | 30.4 | 68.9 |
| Number of runs below loss 30 | 0 | **14** | **14** | 0 |
| Number of runs below loss 50 | 0 | **27** | **27** | 8 |
| Total runs | 60 | 60 | 60 | 60 |

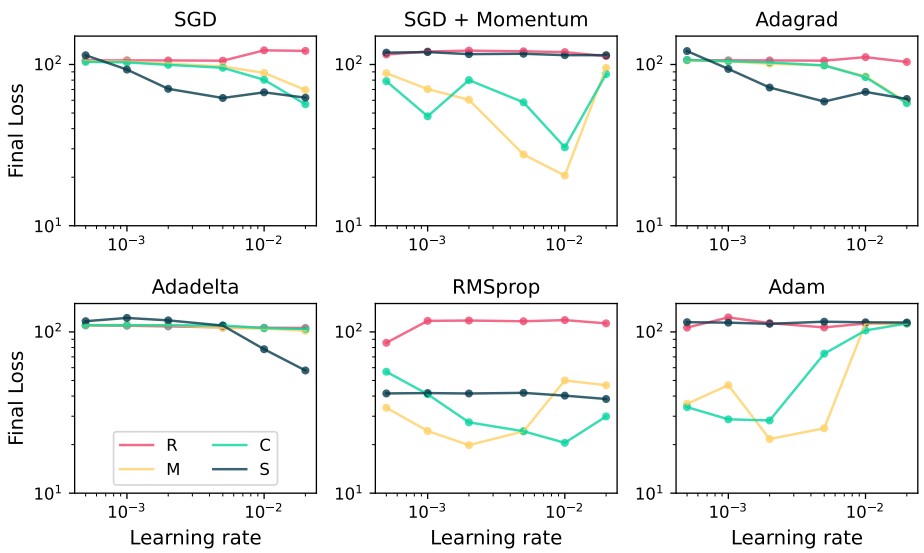

Figure 11: Quantum control, hyperparameter study with different learning rates and optimizers.

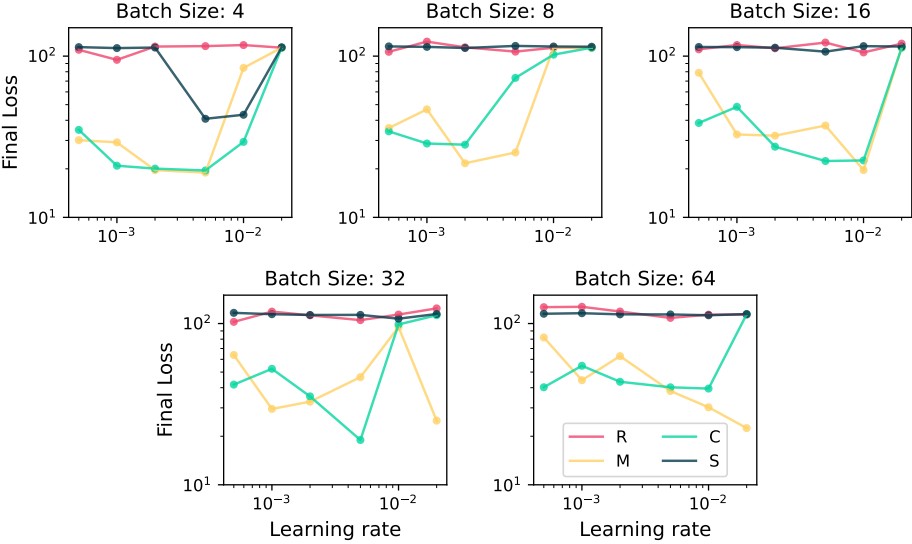

Figure 12: Quantum control, hyperparameter study with different learning rates and batch sizes.

## C  FURTHER VISUALIZATIONS

Here we present further visualizations of optimizations landscapes, the gradient, and our modified update for control tasks different from our toy problem in the main part. These visualizations highlight that many of our observations regarding the modified updates carry over to more complicated settings.

### C.1  LINEAR QUADRATIC REGULATOR

A common control problem is the linear quadratic regular (LQR), which following our Notation in 1 and using constant square matrices $A, B$ can be defined as follows:

$$N(x_i, \theta) = \theta x_i$$
$$S(x_i, c_i) = Ax_i + Bc_i \tag{14}$$

Gradient Descent is known to converge to a global minimum in this optimization task, for appropriate $A$ and $B$ (Fazel et al., 2018). We do not offer a similar mathematical proof here as this typically requires a unique minimum or a loss function, which is then shown to iteratively decrease. As LQR is a non-convex problem and as a distinctive feature of our method is the absence of a loss function, established proof techniques or their ideas do not carry over. Instead, we present a visualization of the flow fields of the gradient field and our modified field similar to those of the toy example, which allows us to assess whether points in the optimization space flow to global minimum. As for visualization, we are restricted to two-dimensional problems, we consider the dynamics in a hyperplane of the full optimization space and we have to choose one with a global minimum. Technically for the restricted case of LQR, the convergence of gradient descent is not guaranteed according to Fazel et al. (2018), so it is also open if the gradient lines end on a global minimum. We choose $n = 5$ time steps, and $A$ and $B$ to be:

$$A = \begin{pmatrix} 0.8 & 0.5 \\ -1.2 & 1.0 \end{pmatrix} \qquad B = \begin{pmatrix} -0.5 \\ -0.6 \end{pmatrix} \tag{15}$$

We visualize the results in Figure 13. Four global minima are located around the points $(-2, 1)$, $(-0.5, -2)$, $(4, 0.5)$ and $(0.5, -6)$. In parts b) and c) of the figure, we observe that every flow line ends on a global minimum. For the minima at $(-2, 1)$ and $(0.5, -6)$, we even see what is typically called an ill-conditioned landscape: it locally resembles a valley with quickly changing loss in one direction and slowly changing in the other. Also noteworthy is that the global minimum at $(-0.5, -2)$ is one of those with a rotating flow field of our modified method in part c). As all of the flow lines of the modified field, just as for the gradient field move toward global minima, we have no obvious evidence for globally suboptimal behavior, indicating the modified field can be used to solve that optimization task-

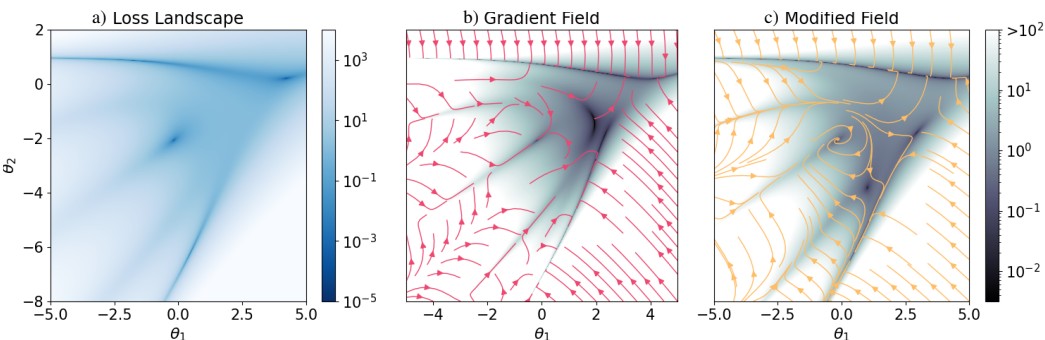

Figure 13: Linear quadratic regulator: a) loss landscape, b) regular gradient field (flow lines in red), c) modified vector field (yellow). For b) and c) the background color shows the length of updates.

## C.2 NEURAL NETWORKS

We present a similar visualization for a network trained on the quantum control task. We again show a two-dimensional subspace of the optimization space and zoom in on key structures that arise:

Figure 14a) shows a sawtooth-like loss landscape with several local minima in the form of multiple dark blue lines. The white areas are local maxima. The gradient field in b) behaves as expected, orthogonal on the lines of equal loss, pointing from local maxima to local minima, and takes on extremely large values. In c), several properties we explained about our modified field can be seen. The flow lines are not orthogonal anymore and have a more balanced scale. Additionally, the updates move straight over the local minima, reducing the chance of getting stuck.

Figure 15a) shows the loss in a different region. A closed manifold of local minima is located in the upper part of the plot, which draws in the large majority of the flow lines in the plot, as shown in b). Moreover, there are saddle points in the lower left part, leading to gradients with large variations in their directions on small scales. In contrast, our modified vector field in c) traverses these local minima and is unaffected by the saddle points.

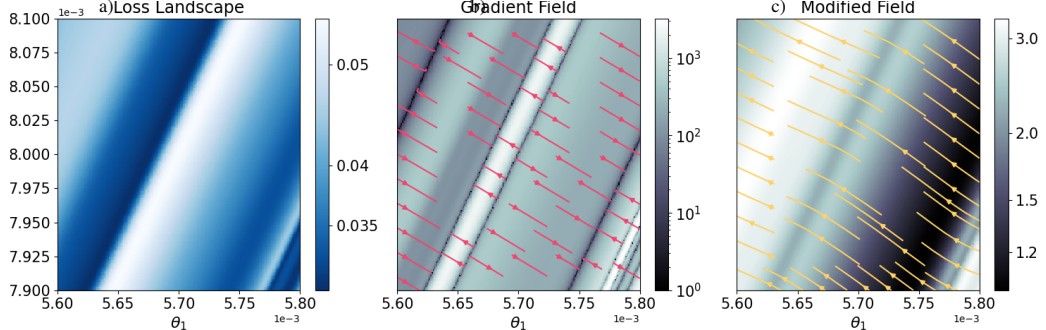

Figure 14: Quantum control task: a) loss landscape, b) regular gradient field (flow lines in red), c) modified vector field (yellow). For b) and c) the background color shows the length of updates.

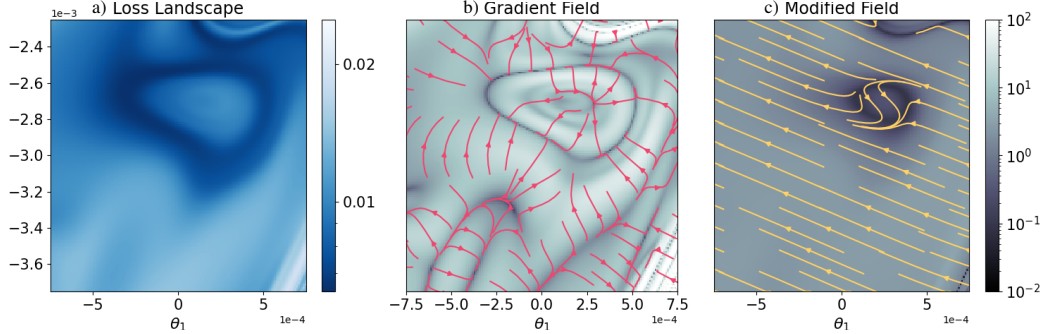

Figure 15: Quantum control task: a) loss landscape, b) regular gradient field (flow lines in red), c) modified vector field (yellow). For b) and c) the background color shows the length of updates.

# D  A SIMPLE CONTACT TASK

Here we present a short discussion of the influence of a more problematic physics solver as those encountered in contact-rich tasks. This addresses the question if our method, which focuses on the gradient flow through the physics simulator, still works.

## D.1  TOY EXAMPLE WITH CONTACT

We start with the changes in the analysis of our toy example. We modify equations 3 by using an absolute value function instead of the identity as an abstraction of a non-differentiable interaction at certain points.

$$
\begin{aligned}
N(x_i, \theta) &= -\theta_1 x_i^2 + \theta_2 x_i \\
S(x_i, c_i) &= |x_i| + c_i
\end{aligned}
\tag{16}
$$

We consider $n = 3$ time steps and visualize the landscape in Figure 17. In the gradient field, we can see that the optimization space is divided by discontinuities originating from the absolute function. One example would be the line from $(-5, 0)$ to $(5, 2)$. Outside these discontinuities, the loss is still a smooth function. So we conclude the differences between gradients and our modified update still hold inside the smooth regions, such as global minima are conserved and local minima or saddle points can vanish. One example would be the saddle point around $(3, 4)$. At the lines of discontinuity, what happens depends on the vector fields on both sides. If they point in the same direction, an optimizer can move through this line without difficulties. If on both sides the points toward the discontinuous connection, this would create structures that an optimizer could get stuck in. For the gradient, one would call this a local minimum and this connects to a widely-accepted observation that contact tasks can give rise to new local minima. We think this argumentation holds equally for the gradient and the modified field and consider it a hint that contact affects both methods in the same way negatively.

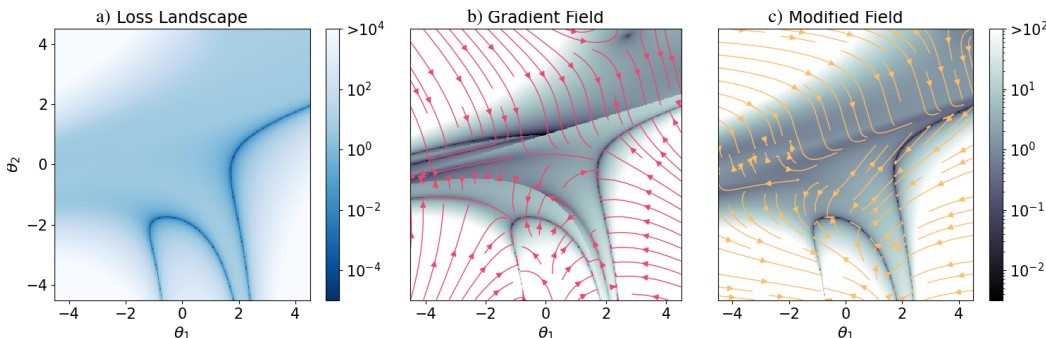

Figure 16: Toy example with contact: a) loss landscape, b) regular gradient field (flow lines in red), c) modified vector field (yellow). For b) and c) the background color shows the length of updates.

## D.2 CART POLE SWING-UP WITH WALLS

We modify our cart pole task to investigate a simple contact problem. With a parameter $w > 0$ introduce two walls, one left at position $-w$ and one right at $w$ that reflect the cart's movement once it hits them. Concretely for the right wall, if the position $x$ of the cart at the end of a time step is $w + d$ with $d > 0$, we instead update the position to $w - d$ and negate the cart's velocity. For the position, this follows the dynamics of the absolute value function considered above, and for the velocity, this is even a discontinuous map.

In our experiments, we used 25 time steps with duration $dt = 0.04$, a continual loss formulation, 4 poles, and wall parameters $w$ from $0.2$ to $1.0$. In this setting, contacts typically happen 4-times per episode for the narrow case $w = 0.2$ and around once for the widest case $w = 1.0$.

The results are shown in 17. We observe in all cases diverging of the stopped update *(S)*. The regular gradient *(R)* is able to solve the task but is outperformed by our modified update *(M)* and our combined update *(C)*. Both of these are able to solve the task equally well.

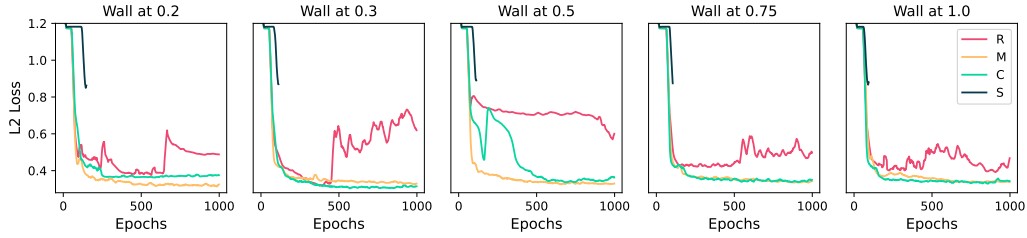

Figure 17: Cart pole with contact for different wall parameters $0.2$, $0.3$, $0.5$, $0.75$ and $1.0$. *(C)* and *(M)* outperform *(R)*. *(S)* diverges. Curves were smoothed over 20 epochs for clarity.

