# OpenReview forum: "Stabilizing Backpropagation Through Time to Learn Complex Physics"
_ICLR.cc/2024/Conference — ICLR 2024 poster_

### Official Review · Reviewer_Djj8 · 2023-10-25

**Soundness:** 2 fair
**Presentation:** 3 good
**Contribution:** 2 fair
**Rating:** 3
**Confidence:** 4

**Summary:**

This paper proposes two simple gradient manipulation strategies to stabilize control learning via backpropagation through time in physics simulation. The approaches are motivated by the fact that naively executing back propagation through time along a long horizon in physics simulation with neural network feedback policy results in gradient exploding and vanishing problems. The first proposed approach cuts the gradients of the neural network inference step and the computed pseudo-gradients have the potential problem of non-rotational-free. The second proposed approach alleviates the issue of the first approach by zeroing out the gradient components with wrong signs. The proposed approaches are evaluated in three dynamics control problems and are shown to outperform the optimization using regular gradients.

**Strengths:**

1. The proposed gradient modification approaches are simple and concise. The approach should be easy to implement.

2. The proposed approach shows great and consistent performances on the designed problems.

**Weaknesses:**

1. Lack of theoretical guarantee for the proposed approach.

2. The approach is only validated on the problems with simple dynamics. Its generalizability to more complex tasks is unknown.

**Questions:**

1. What does the color in Figure 1(b) and 1(c) mean? Does it mean the norm of the gradient?

2. $\theta$ in Eq. (1) is not defined.

3. For the gradient stopping approach (section 3.1), does it guarantee the modified gradient is zero at global minimal?

4. The underlining assumption of the approaches is “most physics simulators come with a well-behaved gradient flow”, which is usually not true in most robotics problems especially when contacts happen [1, 2]. In those cases, the gradients from the simulation usually have much larger gradients than the gradients from the neural network. It would make the proposed approach much stronger if it could be evaluated on more complex robotic tasks such as the ones in [1, 2].

5. Is there any theoretical guarantee of the correctness of the combination approach (section 3.3). It would be helpful to have two proofs: (1) the modified gradients keep the global optimality of the problem, (2) the modified gradient field is rotation-free.

6. The complex version of the cart pole swing-up task is a bit strange in that the network is trained to control multiple poles with the same mechanism. Does it mean the network outputs 4x the number of actions to control 4-pole concurrently? The problem complexity does not indeed increase since basically, four copies of the single-pole policy should work well in this 4-pole task. A more meaningful task would be controlling a 4-linkage pole (i.e. multi-pendulum) which has much more complex dynamics than a single-linkage cart pole.


[1] C. Daniel Freeman, Erik Frey, Anton Raichuk, Sertan Girgin, Igor Mordatch, and Olivier Bachem. Brax - a differentiable physics engine for large scale rigid body simulation.

[2] Jie Xu, Viktor Makoviychuk, Yashraj Narang, Fabio Ramos, Wojciech Matusik, Animesh Garg, Miles Macklin. Accelerated Policy Learning with Parallel Differentiable Simulation

---

> ### Author Response · Authors · 2023-11-20
> **Response to Reviewer Djj8 -- Part 1**
>
> Dear Reviewer,
>
> Thank you for taking the time and sharing your opinion on our work.
>
> We would like to provide some background on our choice of experiments before we answer your questions.
> One aim of this submission is, among others, to start building a link between reinforcement learning and control within scientific computing, two fields with many common questions but, in our perception, largely separated communities. As such, we chose the cart pole and the quantum system as representatives of each field. Both are actually widely used benchmark problems.
> Admittedly, the guidance-by-repulsion system is less widely used, but is actually not as simple as it seems: one of the interaction terms is a quickly increasing repulsion force once drivers and evaders come close to each other, resembling contact dynamics. You are right that our setups could be more complex, but starting with established baseline problems instead of complicated settings was a deliberate choice.
> In that way, our results are as little as possible distorted by secondary issues in order to not draw the focus away from the central topic of our paper, i.e. Backpropagation Through Time.
> Nevertheless, we respect the desire to experiment
> with complex tasks and do not deny the importance of such an evaluation. We personally are excited to address complex environments in future projects.
>
> **Q1) Do the colors indicate the norm of the update in Figure 1b and c?**
>
> Yes, black and white colors indicate the norm of the update. The red and yellow colors of the flow lines indicate the method, using the same color scheme as later in the paper (red: regular gradient, yellow: modified update, green: combined update.
>
> **Q2) $\theta$ not defined in equation (1)**
>
> Apologies. $\Theta$ denotes the network parameters. Thank you for pointing this out.
>
> **Q3) Is it guaranteed that the modified update it zero at global minima?**
>
> Yes. In equation (4), global minima fulfill $F=y$ and therefore the gradient and the modified update are both $0$. It is one of the key properties why we can use this modified vector field to solve the optimization task.

---

> ### Author Response · Authors · 2023-11-20
> **Response to Reviewer Djj8 -- Part 2**
>
> **Q4) "The underlining assumption of the approaches is “most physics simulators come with a well-behaved gradient flow”, which is usually not true in most robotics problems especially when contacts happen [1, 2]. In those cases, the gradients from the simulation usually have much larger gradients than the gradients from the neural network. It would make the proposed approach much stronger if it could be evaluated on more complex robotic tasks such as the ones in [1, 2]."**
>
> The assumption is based on physical intuition. Due to the diverse nature of physical processes and also the many ways they are mathematically modeled, it is not really possible to support such a statement with a generic mathematical proof. We think that conserved quantities limit the extent to which a physical state can change over time. For instance, in a mechanical system with finite energy, things will not move infinitely fast. In our quantum system, the state corresponds to a probability distribution that cannot grow larger than $1$. Even in a contact example, there are restrictions: a ball hitting an elastic wall will change the direction of its velocity abruptly, but the norm of the velocity remains the same. Nevertheless, we understand your wish for further evaluation with a contact example. As pointed out above, we consider the guidance-by-repulsion system a contact example, where discontinuous contact has been replaced by a differentiable relaxation. We also conducted new experiments without such a differentiable relaxation on a modified cart pole system that includes a left and a right wall to restrict the cart's movement. When the cart hits any of the walls, its velocity abruptly changes its sign. We observe that our conclusions still hold in this setting: the modified and combined updates both outperform the regular gradient. We will include these new results and the corresponding plots in the upcoming update of our submission.
>
> **Q5) "Is there any theoretical guarantee of the correctness of the combination approach (section 3.3). It would be helpful to have two proofs: (1) the modified gradients keep the global optimality of the problem, (2) the modified gradient field is rotation-free."**
>
> (1) is true because of the explanation in Q3 (or subsection 3.1 in our paper). Regarding (2), you probably meant the combined vector field here. The statement is not true. One way to see it is by looking at Figure 2 from which one can see the combined vector field can be a non-differentiable field for which rotation is not defined. We think rotation has two effects, a positive and a negative. The positive one is that because of it flow lines need not be perpendicular on the potential lines of the loss, which replaces the extremes in magnitude and direction by a more uniform vector field. The negative one is that it can make it difficult to approach a minimum with an optimizer. It is probably even necessary to use a non-differentiable function to combine these two to achieve a beneficial compromise between the two.
>
>
> **Q6) Clarifications on the cart pole task**
>
> In the 4-pole cart pole task, there is one controller or action output for all four poles. As you correctly described, if there were 4 controllers, one for each pole, this would not increase the difficulty of the task. Our task consists of 4 poles attached to a single cart and with the goal of swinging them all up by moving the cart left and right. As the poles are attached at a single point, we found that repeating the cart for visualization provided most information about the movement of the cart, but we will work on improving the visualization to make sure it is not misleading.

---

> ### Comment · Reviewer_Djj8 · 2023-11-21
> **Reply to Authors' Response**
>
> Appreciate your detailed response and thank you for providing clarifications on the cartpole example. Here are my additional thoughts after reading the response:
>
> 1. Sorry that my question Q3 regarding zero gradients of the modified gradients at global optimal might be confusing. It is certain that if the objective of the problem is to match the target state (i.e. $L = \|F - y\|^2$) the modified gradient is zero at the global minimal. But my question is more on a general setting where we have a general differentiable loss function $L$ that is not in a particular form as state matching.
>
> 2. The dynamics changing of the physics simulation can indeed be very significant and result in gradient explosion issues. This is because of two main reasons:
>
>     a) Contact dynamics can be discontinuous. In the ball hitting wall case given by the authors, though the magnitude of the velocity does not change if the collision is elastic, the direction of the velocity changes abruptly/discontinuously, which can result in an extremely large gradient along the ball reflection axis.
>
>     b) I assume in this work the authors are considering the problem in the context of simulation. Though some physics in many cases is inherently conserved and stable by physics laws, it is not necessarily the case in simulation due to the approximation that has to be made during implementing a simulator. For example, contact dynamics are usually approximated by solving a linear complementarity problem (LCP) which needs to be numerically solved by a solver, and physics is integrated via discrete time steps, both of which can exhibit extra discontinuities (as shown in [1]).
>
>     Because of such complexities of physics simulation, previous papers have proposed different ways to improve policy training in differentiable physics simulation [1, 2, 3]. On the other side, the modified gradients approach shares some similarities with optimizing an open loop action sequence, and previous work (such as [4]) has shown that directly optimizing an open loop action sequence for a complex robotics problem can easily get trapped into local minima. So it would be super valuable and more convincing to evaluate the proposed approach (both the modified gradients version and the combined approach) on complex robotic tasks defined by those previous works. Given the conciseness of the proposed approach, I believe it should not be an extreme effort to implement the approach in the released codebase of previous work (such as in [2] and [4]).
>
> [1] Suh et al., "Do Differentiable Simulators Give Better Policy Gradients?" ICML 2022
>
> [2] Xu et al. Accelerated Policy Learning with Parallel Differentiable Simulation, ICLR 2022
>
> [3] Qiao et al. Efficient Differentiable Simulation of Articulated Bodies, ICML 2021
>
> [4] Freeman et al. Brax - a differentiable physics engine for large scale rigid body simulation

---

> > ### Author Response · Authors · 2023-11-22
> > **Response**
> >
> > Thank you for your interest; we are happy to continue the discussion.
> >
> > **1:**
> > We would argue yes. Generally, a differentiable loss $L$ with a global minima can always be Taylor-approximated around the global minima and then give a quadratic loss locally. Technically it does not have to be a quadratic minimum but could also be a higher-order one, but then the argumentation is still essentially the same, with additional higher-order terms. This is all under the assumption that you have an expression $L(F(x))$ that does not change the loss derivatives $\partial L/\partial F$ but only $\partial F/\partial x$. In this way, they are combined with matrix multiplications according to the chain rule, so it is still reasonable to call this backpropagation.
> >
> > **2a:**
> > You are right that contact need not be continuous. We should have provided some more explanations here. It was meant as an argument that even though contact can be discontinuous, it follows certain physical rules, and will not lead to states growing indefinitely.
> > However, as you point out, a backpropagated feedback vector still can have a huge derivative, and if enough contact events happen this can lead to an explosion. Thank you for clarifying this issue.
> >
> > **2b:**
> >  Discretization of differential operations leads to numerical errors, we have no disagreements about this. However, suppressing those as best as possible is the goal of any discretization scheme. Again our quantum experiment, which by the way includes one linear solve in each of the 100 time steps, is a good example because preserving the total probability (the L2 norm) is the top feature of any quantum simulator. If not present, any control task could just be solved by pumping probability in and out of the system in an unphysical way. However, your explanation made us realize that this might be harder to achieve in a contact scenario.
> >
> > You are right that implementing our technique is not an extreme effort, but ensuring that our approach is implemented correctly and obtaining stable results that are not invalidated by a suboptimal hyperparameter at the same time isn't a trivial task and we're hesitant to provide hastened or potentially misleading results. We will nonetheless continue our efforts in this area to add such an experiment in a future version of our submission.
> >
> > Thus, it is a valid point that contact introduces substantial difficulties for learning tasks, and we have not studied contact scenarios in detail. In the update of our paper, we will make this limitation clear, and we believe it will be a very interesting avenue of future work to explore how well our methods work on contact scenarios that go beyond the relatively simple contact and wall collisions that we have added during the rebuttal. These show that our method still works in simple collision scenarios, but this is of course no proof that the positive effects carry over to new scenarios in the same way.

---

### Official Review · Reviewer_BicD · 2023-10-29

**Soundness:** 3 good
**Presentation:** 3 good
**Contribution:** 3 good
**Rating:** 8
**Confidence:** 4

**Summary:**

This paper tries to tackle the gradient explosion / vanishing problem in backpropagation through time. The authors first propose a method to stop the gradient of the feedback policy at each step with respect to the state while keeping the rest of the states active. Then, the authors point out that this update has problems due to rotation, and proposes to combine the original gradient and the modified gradient to tackle how rotation can slow down convergence of gradient-based methods. The authors compare their method against regular gradient descent and stopped gradient descent, and show that the method performs better.

**Strengths:**

1. The visualizations of the problems are well done and the problems that the authors are trying to convey are clearly communicated to the reader.
2. The method shows convincing improvement over competitors on simple experiments.

**Weaknesses:**

(1) I have spent some time trying to understand the authors' argument on why it would be beneficial to stop the gradient of the policy with respect to the current state (i.e. $\partial_x N$ in author's notation). But I still find that the motivations and the justifications for this is quite weak.

Fundamental theorem of calculus (generalized Stokes) tells us a nice connection about the loss and its gradient, so for smooth systems that the authors are considering, if we integrate back the gradient, we should get the loss. So when the authors set some of the terms to zero, there must be some surrogate loss that the modified gradient is considering. I'm willing to agree with the authors that this loss might have a better landscape compared to the original one; but how do we know that whether or not this landscape is completely unrepresentative or the original one?

I think this is the most important section and contribution of the paper that is relatively not very well motivated in the paper, and am willing to give the paper a much better score if the authors can be more convincing about these points with some theory to back it up.

(2) Related to above points, in the experiment results, the performance of combined (C) is much better than the Modifed (M). But it's not clear if this is because rotation is fixed, or if modified is simply not good because of the above issue.

(3) The authors are missing a large branch of relevant work on studying the efficacy of gradient-based methods in the setting of optimization through differentiable simulation [1,2,3,4], a lot of the issues that the authors have mentioned for Back Propagation Through Time (BPTT) could have benefited from citing these works. These works also have existing methods for improving the performance of BPTT (e.g. total propagation from [1], alpha-order gradients from [3]), which would have made stronger baselines.

(4) It seems to me like the authors are mainly considering the difficulty of BPTT as considering complicating feedback from control actions, but previous works have mainly motivated the shortcomings of BPTT through the lens of characteristics in the dynamics such as chaos [1,2,3] or discontinuities through contact [3,4]. It is unclear if the author's method will improve performance for these difficult systems.

(5) This is minor, but I wished the authors used a more standard notation from nonlinear control (where state-actions are (x,u), dynamics are f, and policy is k) or Reinforcement Learning (states-actions are (s,a) dynamics can still be f, policy is $\pi$).


[1] Parmas et al., "PIPPS: Flexible Model-Based Policy Search Robust to the Curse of Chaos", ICML 2018

[2] Metz et al., "Gradients are not all you need"

[3] Suh et al., "Do Differentiable Simulators Give Better Policy Gradients?" ICML 2022

[4] Antonova et al., "Rethinking Optimization with Differentiable Simulation from a Global Perspective", CoRL 2022

**Questions:**

(1) I think the biggest question is: why is stopping the gradient $\partial_x N$ more beneficial, and how do we ensure it's taking more globally beneficial steps compared to the actual gradient? How do we know it's not completely wrong by ignoring these terms, or if there is some pathological system where throwing away these terms will completely make the optimization fail?

A convincing case that the authors could consider is the Linear Quadratic Regulator problem where we have linear dynamics $S(x,c) = ax + bc$ and a linear policy $N(x,\theta)=\theta x$ over some horizon. It is known that gradient descent will converge to the optimal parameters $\theta$ for this problem [5]. But if we assume that $\partial_x N = 0$, can we actually converge to the minima of the original problem at all?

[5] Fazel et al., "Global Convergence of Policy Gradient Methods for the Linear Quadratic Regulator", ICML 2018

---

> ### Author Response · Authors · 2023-11-20
> **Response to Reviewer BicD -- Part 1**
>
> Dear Reviewer,
>
> Thank you for detailed feedback; it helps us a lot to improve the future version of our paper. Below we provide additional details to address your concerns.
>
> **W1) "I have spent some time trying to understand the authors' argument on why it would be beneficial to stop the gradient of the policy with respect to the current state (i.e. $\partial_x N$ in author's notation). But I still find that the motivations and the justifications for this is quite weak."**
>
> First, regarding your comment “So when the authors set some of the terms to zero, there must be some surrogate loss that the modified gradient is considering.” We'd like to point out that this statement is not true: Not every vector field is a gradient field, and a generic modification to a gradient field will in most cases lead to this situation. A minimal example with a gradient stop and two variables $(x,y)$ is the following:
>
> $a(x,y)=-2xy$
>
> $b(x,y)= xy$
>
> $c(a,b) = a+b$
>
> $c(a(x,y),b(x,y)) = -xy$
>
> The gradient of $c$ w.r.t $(x,y)$ is $(-y,-x)$. The potential field it came from is $-xy$.
> A gradient stop on the second argument of $a$ sets $\partial a / \partial y=0$ instead of $-2x$ and modifies the vector field:
>
> $\frac{dc}{dy}=\frac{\partial c}{\partial a} \frac{da}{dy}+\frac{\partial c}{\partial b} \frac{db}{dy} = 1 \cdot 0 + 1 \cdot x = x$
>
> The modified vector field is then $(-y, x)$, the classical example of a rotating vector field that is not a gradient field to any potential (or loss, in optimization language).
>
> The absence of a surrogate loss that such a modified vector field corresponds to has several implications on the analysis of this technique. Among others, instead of talking about minima of a loss we are forced to talk about critical or fix-points of a vector field. You also asked if this new field is representative of the original loss. The answer is yes in the sense that, as argued in the paper, global minima of the original loss are fix-points of the modified vector field. Other fix-points of the gradient field such as saddle points or local minima do not have to be fix-points of the modified vector field. This is a good property as those are widely considered to be obstacles during optimization.
>
> These are local arguments and your concern about the global behavior (Question 1) is justified. Your suggestion to analyze the situation for a linear quadratic regulator (LQR) is a great idea. Most convergence  proofs include an estimate of how one optimization iteration decreases the loss or the distance to a unique minimizer. However, these ideas do not carry over to our case as we do not have a surrogate loss and LQR is non-convex. Thus, we believe our results indicate that developing this theory will be a very interesting but also challenging topic for future work.
>
> We can however empirically evaluate the gradient and the modified field with visualizations. While we have presented such a visualization for the toy example in our submission, we have now extended this visualization to LQR and the neural networks trained in our experiments. As these are not two-dimensional optimization problems, our visualization uses two-dimensional subspaces of the optimization space and therefore depicts the dynamics constrained to this subspace. These plots highlight that central aspects of our method carry over from the toy problem to practical scenarios: i) global minima remain attractors in the modified vector field ii) the gradient field has many saddle points with chaotically changing flow lines around them while the modified vector field does not iii) local minima in the gradient fields are not attractors of the modified vector field. We will include these plots and a discussion in our updated version.
>
> **W2) "Related to above points, in the experiment results, the performance of combined (C) is much better than the Modifed (M). But it's not clear if this is because rotation is fixed, or if modified is simply not good because of the above issue."**
>
> If we understand this point correctly, you are worried this is due to (C) being closer to the gradient (R) than (M) is to (R) (and assuming (R) is good). A counterargument to that hypothesis is that we observed many runs with (M) being better than (R); the quantum system is the best example where this was consistently the case. Nevertheless, we share your curiosity to gain more insights about the rotational components for the actual experiments. Unfortunately, rotation is a second-order quantity and in a large-parameter setting the Hessian matrix is not efficiently accessible.

---

> ### Author Response · Authors · 2023-11-20
> **Response to Reviewer BicD -- Part 2**
>
> **W3) "The authors are missing a large branch of relevant work on studying the efficacy of gradient-based methods in the setting of optimization through differentiable simulation [1,2,3,4], a lot of the issues that the authors have mentioned for Back Propagation Through Time (BPTT) could have benefited from citing these works. These works also have existing methods for improving the performance of BPTT (e.g. total propagation from [1], alpha-order gradients from [3]), which would have made stronger baselines."**
>
> We agree that our related work should be expanded, and we will add a more extensive discussion of these topics in the appendix. Regarding the two suggestions for additional baselines, we think both of them are largely orthogonal to our work.
> - We think that contact equally presents challenges to the regular gradient field and our modified field. A question by another reviewer was about the effects of contact in our analysis and the gradient field and the modified field. The bottom line was the optimization space splits into several parts with the smoothness assumption still fulfilled within each part (here the positive aspects of our method still hold) but where two such parts intersect, problematic structures arise when the vector fields point in different directions, e.g. fixpoints that correspond to local minima in the original loss. Since both the regular gradient and our modified field can be equally affected by this, we are not sure if a technique designed to improve optimization in contact tasks by switching to a zero-order quantity but only applied to one of the vector fields in question would really be a fair comparison.
> - Similarly, 'Total backpropagation' is a way of combining two different estimators for a gradient when the transition from one state $x_i$ to the next $x_{i+1}$ is stochastic. As in our case both controller/policy and dynamics/model are deterministic, all three estimators end up being the same. Technically one could argue what you asked for is already included as a baseline in form of the regular gradient. The reason why we kept stochasticity out of the picture in our paper is that the underlying problem we study and our proposed way to address it do not include any stochastic effects; including a probabilistic argumentation would be an interesting direction, but one that is not directly related to the core motivation for our algorithm.
>
> We are thankful for pointing out these related methods. We would consider them not so much being additional baselines, as explained above, but instead rather as something that can be combined with our method by using our modified/combined update where they use the first-order gradient. Seeing if they work together is an interesting question and we hope that we can explore this in a future project.

---

> ### Author Response · Authors · 2023-11-20
> **Response to Reviewer BicD -- Part 3**
>
> **W4) "It seems to me like the authors are mainly considering the difficulty of BPTT as considering complicating feedback from control actions, but previous works have mainly motivated the shortcomings of BPTT through the lens of characteristics in the dynamics such as chaos [1,2,3] or discontinuities through contact [3,4]. It is unclear if the author's method will improve performance for these difficult systems."**
>
> Regarding chaotic dynamics, we think all of our experiments fulfill this criterion. One of our examples (cart pole swing-up) is in fact the same system as in [1], and their other two experiments are to learn a linearization around an unstable fix point (cart pole and unicycle balancing). We are not sure if these systems are really more difficult than ours. What might be helpful here is to mention that the authors are not talking about chaotic dynamics of the physical system but of the optimization dynamics, and then, as many others, the connections to gradient sizes. You are right that we argued more around the feedback from the control actions as we think there is room for improvement best seen in our figure 5e for the quantum system: it has no chaotic physics and no contact dynamics, and yet the massive gradient sizes are completely gone just by removing the undesirable feedback from the control actions.
>
> Regarding systems with contact, we think that our guidance-by-repulsion system falls in that category as it includes a repulsion term that quickly grows when drivers and evaders come near each other, resembling a differential relaxation of discontinuity through contact. In addition, we conducted new experiments with a modified cart pole system that includes walls abruptly changing the direction of the cart once it touches them. We varied the position of the walls to vary the number of contacts happening and in all cases, the modified and combined updates were better than the regular update. The plots will be included in our updated version of the paper.
>
>
> **W5) "This is minor, but I wished the authors used a more standard notation from nonlinear control (where state-actions are (x,u), dynamics are f, and policy is k) or Reinforcement Learning (states-actions are (s,a) dynamics can still be f, policy is ). "**
>
> We focused especially on the notation for the gradient stop and the expressions around it as we assumed this part could be most easily misunderstood, and less on the letters themselves. We apologize if this has been a source of confusion.

---

> ### Comment · Reviewer_BicD · 2023-11-23
> **Comment**
>
> I would like to thank the authors for their response and hard work. I think I was misreading some of the arguments in the paper, and the authors have done a good job of explaining the key thing I was a bit mislead about: the vector fields caused by truncated gradients conserve the critical points of the original optimization problem.
>
> With new convincing experiments on LQR and the setting of contact, I am happy to update the score to 8. My final suggestion would be that the authors should make an effort to make the point in 3.1 much more clear and evident. Like reviewer Djj8 suggested, I think it would be good if the authors can explicitly note this as a theorem / proposition so that the readers can immediately realizes the validity of this technique.

---

> > ### Comment · Reviewer_BicD · 2023-11-23
> > **Cart-pole contact plot**
> >
> > Also in the cart-pole with contact plot in D.2, I expected the different methods to have names CMRS but the legends say FPCS, likely a typo?

---

> > > ### Author Response · Authors · 2023-11-23
> > > **Response**
> > >
> > > Thank you for your response. Yes, there is a typo in Figure 17: F should have been R, and P should have been M. We will definitely fix this.

---

### Official Review · Reviewer_Z3pv · 2023-10-29

**Soundness:** 4 excellent
**Presentation:** 4 excellent
**Contribution:** 3 good
**Rating:** 8
**Confidence:** 2

**Summary:**

The paper studies the gradient exploding and vanishing problem due to recurrent operations in the context of control optimization with differentiable physics in the loop. It constructed a toy example to illustrate the problem and proposed a simple method to modify the gradient such that a better optimization landscape can be obtained.

The method is validated on three tasks and compared to alternative approaches (regular gradients, no combination, no long-range back-propagation).

**Strengths:**

- The paper is well written. The toy example and visuals are very helpful in terms of understanding the problem and solution. The choice of the example is well explained.
- The proposed method is conceptually simple (gradient stopping and sign check) but sheds light on using gradient modification to stabilize optimization.

**Weaknesses:**

**Experiments**
- The visualization of results could be improved. For example in Fig 3, multiple curves have the same color and overlap each other, making it difficult to draw a clear conclusion. It might be better to draw a mean-std plot for each method where the std (computed over trials) is shaded.

**Questions:**

1. The analysis assumes a simplified simulator (identity mapping + control). However, the transition between steps could be complex (e.g., non-linear contact) but the controller is simple (PD controller). How does this change the analysis? One concrete example is robot control.
2. In the cart-pole experiment, Fig 4 suggests optimization is less stable with fewer poles. Is there an explanation for why this happens?

---

> ### Author Response · Authors · 2023-11-20
> **Response to Reviewer Z3pv**
>
> Dear Reviewer,
>
> Thank you for your interest and for sharing your thoughts.
>
> **Q1) How does our analysis change when we consider a more complex transition between states, i.e. instead of identity use a non-linear contact?**
>
> In the analysis with our toy example, let us assume we replace the identity mapping in the physics path with an absolute value function. This means there is one point (at 0) with an abrupt change of behavior, serving as an abstraction of a contact interaction. For optimization parameters with corresponding trajectories that do not touch this non-differentiable point, our analysis still holds. Effectively, the optimization space is divided into several parts where the smoothness assumption still holds and the positive effects of our modification of the backpropagation path still set in, for instance, global minima are conserved. At the intersection of two such parts, two things can happen: i) The vector field on each side points in the same direction, and although this is not a mathematically smooth transition, it is unproblematic in terms of the optimization dynamics, since optimization parameters when moving from one part into the other continue moving in the same way. ii) The vector fields on each side point in opposite directions, i.e. towards each other. This slows down the optimization or can even create structures akin to local minima of the original loss.
>
> For an illustration, we also regenerated the toy example plot with an absolute value function instead of the identity function, as mentioned above. We observe basically what we described above; some problematic structures emerge at the intersections, but overall the modified field has more balanced updates and better resembles the original loss landscape. We will include this plot in our update.
>
> There are two more things that we would like to mention about this argumentation:
> a) Even when using a differentiable relaxation of a point contact, the above argumentation still holds. A good surrogate model for contact should never change the dynamics of the rest of the system and therefore, all that should happen in the optimization plot is a smoothing of non-differentiable lines. b) That contact interactions can create new local minima is not a specific disadvantage of our method. The above argumentation for why this can happen also holds for the gradient field used by all other methods and connects to a widely accepted observation that contact dynamics can lead to new local minima.
>
> Please also note that the guidance-by-repulsion model has strong repulsive terms quickly becoming large when drivers or evaders get near to each other or themselves. This resembles contact interactions.
> To provide additional support for the intuition that our method applies to contact scenarios, we have run additional experiments where we studied a modified cart pole problem including left and right walls that restrict the cart's movement. When the cart hits one of them, its velocity is reflected. We can report that our conclusions still hold in this setting: the modified and combined updates both outperform the regular gradient. We will include these new results and corresponding plots in the upcoming update of our submission.
>
> **Q2) Why is optimization less stable with fewer poles in Figure 4, cart pole?**
>
> This is a good observation. In our paper, we argue that complexity matters when evaluating our method. I.e., whether trading the imbalanced gradient updates for a more balanced vector field with rotation pays off. Our explanation here is that for the easiest task, the gradient imbalance is still manageable and that it does not yet outweigh the complications of a rotational vector field.
>
> **Remark about plotting style**
>
> We actually experimented with this initially. The reason we decided against a mean and standard deviation plot style is that in recurrent training setups, the learning curves can look quite unstable and we felt this was better reflected with three individual learning curves than averaged out in a mean and standard deviation plot. Nevertheless, we will provide the mean and standard deviation plots in the appendix of our updated submission in addition to the ones in the main part.

---

### Official Review · Reviewer_kKUm · 2023-11-01

**Soundness:** 3 good
**Presentation:** 4 excellent
**Contribution:** 3 good
**Rating:** 8
**Confidence:** 3

**Summary:**

An learning algorithm is proposed for neural network systems that interact with a differentiable simulator. The idea is to train the neural network policy network with gradient descent, except to stop the gradient from backpropagating through the network multiple times. It is argued that backpropagating through time leads to optimization difficulties for gradient methods (e.g. the well-known gradient explosion in recurrent neural networks). Stopping the gradient in this way doesn't change the objective function, but can improve the dynamics of learning. The authors present nice simulated examples and a series of three experiments

**Strengths:**

- Very well written. Excellent presentation with nice examples.
- The toy examples and figures really helped with illustrating the points.
- Experiments are clear and support the claims.
- Differentiable control of simulators is a topic of interest. This could have high impact.

**Weaknesses:**

- I think the paper could have benefited from having more discussion of the chosen experiment applications. Knowing how these applications compare to potential real-world applications in terms of complexity would have been valuable context for the section on computational cost. It also would have given some context on how well this method might scale to complicated simulations.

**Questions:**

None

---

> ### Author Response · Authors · 2023-11-20
> **Response to Reviewer kKUm**
>
> Dear Reviewer,
>
> Thank you for your time and for your encouraging comments.
>
> **Question) More discussion of the applications of the chosen experiments**
>
> To provide additional background about our choice of experiments, we primarily selected them to bridge the communities of reinforcement learning and control within scientific computing. We believe these are two highly interesting fields with many common questions but two, in our perception, largely separated communities. As such, we chose the cart pole and the quantum system as representatives of each field. Both are actually widely used benchmark problems. Admittedly, the guidance-by-repulsion system is less widely used, but is actually not as simple as it seems: one of the interaction terms is a quickly increasing repulsion force once drivers and evaders come close to each other, resembling contact dynamics.
>
> It is true that our experiments are not yet real-world applications, but we think it is important to first do groundwork by investigating established problems. We agree that real-world use cases are an important goal, and we plan to work on these in future projects.

---

### Author Response · Authors · 2023-11-22
**Response to all Reviewers**

Dear Reviewers,

Thank you again for your feedback. We have focused on including the additional results in the appendix (sections C and D) and will include the additional clarifications and comments from the rebuttal in the main text in the next revision. The two primary updates are:

**1) Contact dynamics:**
 As a new experiment, we investigated the cart pole example once again with two walls restricting the cart's movement and added it to the appendix. In this simple contact scenario, we could observe the positive effects of our method.

**2) Further analysis of our method:**
 Regarding your questions about the observations in our toy example and whether they carry over to more complicated cases, we created similar visualizations for more complicated cases. This includes the suggested linear quadratic regulator and our trained networks. These are included in the appendix and one can observe that our modified field behaves in the same way as we argued in the main part.

We would like to thank you again for your feedback about our submission, The authors

---

### Meta-Review · Area_Chair_Jyht · 2023-12-05

**Metareview:**

This paper proposes a learning algorithm designed to stabilize gradients when backpropagating through time. Most reviewers found the work to be well-written with a simple approach that shows competitive performance on simple benchmarks. The discussion raised some questions about the theoretical guarantees and about how the method would extend to more complex scenarios. In their response, the authors added some new experiments and and analysis that seems to confirm the broader applicability of the results, and therefore I recommend acceptance.

**Justification For Why Not Higher Score:**

The method is largely heuristic, lacking strong theoretical guarantees, and relies on empirical validation for support.

**Justification For Why Not Lower Score:**

Simple idea with good performance, convincing experiments, and clear writing.

---

### Decision · Program_Chairs · 2024-01-16

Accept (poster)